# Managing Temporal Resolution in Continuous Value Estimation: A Fundamental Trade-off

**Zichen Zhang**[*], **Johannes Kirschner, Junxi Zhang, Francesco Zanini, Alex Ayoub,**
**Masood Dehghan, Dale Schuurmans**
University of Alberta
{zichen2,jkirschn,junxi3,fzanini,aayoub,masood1,daes}@ualberta.ca

## Abstract

A default assumption in reinforcement learning (RL) and optimal control is that observations arrive at discrete time points on a fixed clock cycle. Yet, many applications involve continuous-time systems where the time discretization, in principle, can be managed. The impact of time discretization on RL methods has not been fully characterized in existing theory, but a more detailed analysis of its effect could reveal opportunities for improving data-efficiency. We address this gap by analyzing Monte-Carlo policy evaluation for LQR systems and uncover a fundamental trade-off between approximation and statistical error in value estimation. Importantly, these two errors behave differently to time discretization, leading to an optimal choice of temporal resolution for a given data budget. These findings show that managing the temporal resolution can provably improve policy evaluation efficiency in LQR systems with finite data. Empirically, we demonstrate the trade-off in numerical simulations of LQR instances and standard RL benchmarks for non-linear continuous control.

## 1 Introduction

In many real-world applications of control and reinforcement learning, the underlying system evolves continuously in time [Eliasmith and Furlong, 2022]. For instance, a physical system like a robot is naturally modelled as a stochastic dynamical system. Nonetheless, sensor measurements are typically captured at discrete time intervals, which entails choosing the sampling frequency or measurement *step-size*. This step-size is usually treated as an immutable quantity based on prior measurement design, but it has a significant impact on data efficiency [Burns et al., 2023]. We will see that, from a data-cost perspective, learning can be far more data efficient if it operates at a temporal resolution that is allowed to differ from a prior step-size choice.

In this work, we investigate episodic policy evaluation with a finite data budget to provide a key initial step to addressing broader research questions on the impact of temporal resolution in reinforcement learning. We show that data efficiency can be significantly improved by leveraging a precise understanding of the trade-off between approximation error and statistical estimation error in value estimation — two factors that react differently to the level of temporal discretization. Intuitively, employing a finer temporal resolution leads to a better approximation of the continuous-time system from discrete measurements; however, under a fixed data budget, denser data within each trajectory results in fewer trajectories, leading to increased estimation variance due to system stochasticity. This implies that, for a given data cost, it can be beneficial to increase temporal spacing between recorded data points beyond a pre-set measurement step-size. This holds true for any system with stochastic dynamics, even if the learner has access to *exact* (noiseless) state measurements.

---

[*]All authors contributed equally

The main contributions of this work are twofold. First, we conduct a theoretical analysis focusing on the canonical case of Monte-Carlo value estimation in a Langevin dynamical system (linear dynamics perturbed by a Wiener process) with quadratic instantaneous costs, which corresponds to policy evaluation in linear quadratic control (LQR). To formalize the impact of time discretization on policy evaluation, we present analytical expressions for the mean-squared error that *exactly characterize the approximation-estimation trade-off* with respect to the step-size parameter. From this trade-off, we derive the optimal step-size for a given Langevin system and characterize its dependence on the data budget. Second, we carry out a numerical study that illustrates and confirms the trade-off in both linear and non-linear systems, including several MuJoCo control environments. The latter also highlights the practical impact of the choice of sampling frequency, which significantly affects the MSE, and we therefore provide recommendations to practitioners for properly choosing the step-size parameter.

## 1.1 Related Work

There is a sizable literature on reinforcement learning for continuous-time systems [e.g. Doya, 2000, Lee and Sutton, 2021, Lewis et al., 2012, Bahl et al., 2020, Kim et al., 2021, Yildiz et al., 2021]. These previous works largely focus on deterministic dynamics without investigating trade-offs in temporal discretization. A smaller body of work considers learning continuous-time control under stochastic [Baird, 1994, Bradtke and Duff, 1994, Munos and Bourgine, 1997, Munos, 2006], or bounded [Lutter et al., 2021] perturbations, but their objective is to make standard learning methods more robust to small time scales [Tallec et al., 2019], or develop continuous-time algorithms that unify classical methods in discrete-time [Jia and Zhou, 2022a,b], without explicitly addressing temporal discretization. However, we find that managing temporal resolution offers substantial improvements not captured by previous studies.

The LQR setting is a standard framework in control theory and it gives rise to a fundamental optimal control problem [Lindquist, 1990], which has proven to be a challenging scenario for reinforcement learning algorithms [Tu and Recht, 2019, Krauth et al., 2019]. The stochastic LQR considers linear systems driven by additive Gaussian noise with a quadratic cost, minimised using a feedback controller. Although this is a well-understood scenario with a known optimal controller in closed form [Georgiou and Lindquist, 2013], the statistical properties of the long-term cost have only recently been investigated [Bijl et al., 2016]. Our research closely relates to the now extensive literature on reinforcement learning in LQR systems [e.g. Bradtke, 1992, Krauth et al., 2019, Tu and Recht, 2018, Dean et al., 2020, Tu and Recht, 2019, Dean et al., 2018, Fazel et al., 2018, Gu et al., 2016]. These works uniformly focus on the discrete time setting, although the benefits of managing spatial rather than temporal discretization have also been considered [Sinclair et al., 2019, Cao and Krishnamurthy, 2020]. Wang et al. [2020] studies continuous-time LQR, focusing on the exploration problem. Basei et al. [2022] provides a regret bound depending on sampling frequency, for a specific algorithm based on least-squares estimation. Their analysis considers approximation and estimation errors independently, without identifying a trade-off.

There is compelling empirical evidence that managing temporal resolution can greatly improve learning performance [Lakshminarayanan et al., 2017, Sharma et al., 2017, Huang et al., 2019, Huang and Zhu, 2020, Dabney et al., 2021, Park et al., 2021], typically achieved through options [Sutton et al., 1999], a specific instance of which is action persistence, achieved by maintaining a fixed action over multiple time steps (also known as action repetition). Recently, these empirical findings have been supported by an initial theoretical analysis [Metelli et al., 2020], showing that temporal discretization plays a role in determining the effectiveness of fitted Q-iteration. Their analysis does not consider fully continuous systems, but rather remains anchored in a base-level discretization. Furthermore, it only provides worst-case upper bounds, without capturing detailed practical trade-offs. Lutter et al. [2022] discusses the practical trade-off on time discretization but do not provide theoretical support. Bayraktar and Kara [2023] analyzes a trade-off between sample complexity and approximation error, which however requires the state and action spaces of the diffusion process to be discretized to yield an MDP. The two components in the trade-off are analyzed independently, unlike the unified statistical analysis provided in our work, and no exact characterization is presented.

## 2 Policy Evaluation in Continuous Linear Quadratic Systems

In the classical continuous-time linear quadratic regulator (LQR), a state variable $X(t) \in \mathbb{R}^n$ evolves over time $t \geq 0$ according to the following equation:

$$\mathrm{d}X(t) = \mathbf{A}X(t)\,\mathrm{d}t + \mathbf{B}U(t)\,\mathrm{d}t + \sigma\,\mathrm{d}W(t). \tag{1}$$

The dynamical model is fully specified by the matrices $\mathbf{A} \in \mathbb{R}^{n \times n}$, $\mathbf{B} \in \mathbb{R}^{n \times p}$ and the diffusion coefficient $\sigma$. The control input $U(t) \in \mathbb{R}^p$ is given by a fixed policy, and $W(t)$ is a Wiener process. The state variable $X(t)$ is fully observed. For simplicity, we assume that the dynamics start at $X(0) = \overrightarrow{0} \in \mathbb{R}^n$ [c.f. Abbasi-Yadkori and Szepesvári, 2011, Dean et al., 2020].

The quadratic cost $J$ is defined for positive definite, symmetric matrices $\mathbf{Q} \in \mathbb{R}^{n \times n}$ and $\mathbf{R} \in \mathbb{R}^{p \times p}$, a *system horizon* $0 < \tau \leq \infty$ and a discount factor $\gamma \in (0, 1]$:

$$J_\tau = \int_0^\tau \gamma^t \left[ X(t)^\top \mathbf{Q} X(t) + U(t)^\top \mathbf{R} U(t) \right] dt. \tag{2}$$

In the following, we consider the class of controllers given by static feedback of the state, i.e.: $U(t) = KX(t)$ where $K \in \mathbb{R}^{p \times n}$ is the static control matrix yielding the control input. It is well known that in infinite horizon setting, the optimal control belongs to this class. Given such an input, the LQR in Equation (1) can be further reduced to a linear stochastic dynamical system described by a Langevin equation. Using the definitions $A := \mathbf{A} + \mathbf{B}K$ and $Q := \mathbf{Q} + K^\top RK$, we express both the state dynamics and the cost in a more compact form:

$$dX(t) = AX(t)\, dt + \sigma\, dW(t), \qquad J_\tau = \int_0^\tau \gamma^t X(t)^\top Q X(t)\, dt. \tag{3}$$

The expected cost w.r.t. the Wiener process is $V_\tau = \mathbb{E}[J_\tau]$. The policy plays a role in this work solely from its impact in the closed-loop dynamics $A$. Equation (3) is what we analyze in the following. We explicitly distinguish the *finite-horizon setting* where $\tau < \infty$, $\gamma \leq 1$ and the cost is $V_\tau$, and the *infinite-horizon setting* where $\tau = \infty$, $\gamma < 1$ and the cost is $V_\infty$. In order not to incur infinite costs in either scenario, a stable closed-loop matrix $A$ should be assumed. Note that the existence of a stabilizing controller is guaranteed under the standard controllability assumptions in LQR [Fazel et al., 2018, Abbasi-Yadkori et al., 2019, Dean et al., 2020]. Thus the closed-loop stability can be safely assumed.

**Monte-Carlo Policy Evaluation**    Our main objective in *policy evaluation* is to estimate the expected cost from discrete-time observations. To this end, we choose a uniform discretization of the interval $[0, T]$ with increment $h$, resulting in $N = T/h$ time points $t_k := kh$ for $k \in \{0, 1, \ldots, N\}$. Here, the *estimation horizon $T$*, such that $T < \infty$ and $T \leq \tau$, is chosen by the practitioner (for simplicity assume that $T/h$ is an integer). With the $N$ points sampled from one trajectory, a standard way to approximate the integral in Equation (3) is through the *Riemann sum estimator*

$$\hat{J}(h) = \sum_{k=0}^{N-1} \gamma^{t_k} h X(t_k)^\top Q X(t_k). \tag{4}$$

To estimate $V_\tau$, we average $M$ independent trajectories with cost estimates $\hat{J}_1, \ldots \hat{J}_M$ to obtain the *Monte-Carlo estimator*:

$$\hat{V}_M(h) = \frac{1}{M} \sum_{i=1}^M \hat{J}_i(h) = \frac{1}{M} \sum_{i=1}^M \sum_{k=0}^{N-1} \gamma^{t_k} h X(t_k)^\top Q X(t_k).$$

Our primary goal is to understand the mean-squared error of the Monte-Carlo estimator for a fixed system (specified by $A$, $\sigma$ and $Q$), to inform an optimal choice of the step-size parameter $h$ for a *predetermined data budget $B = M \cdot N$*.

Note that one degree of freedom remains in choosing $M$ and $N$. For simplicity, we require that in the finite-horizon setting, the estimation grid is chosen to cover the full episode $[0, \tau]$ which leads to the constraint $T = \tau = N \cdot h$. We write the mean-squared error-surface as a function of $h$ and $B$:

$$\mathrm{MSE}_T(h, B) = \mathbb{E}\big[(\hat{V}_M(h) - V_T)^2\big]. \tag{5}$$

In the infinite horizon setting, i.e. $\tau = \infty$, the *estimation horizon $T$* is a free variable chosen by the experimenter that determines the number of trajectories through $M = \frac{B}{N} = \frac{Bh}{T}$. The mean-squared error for the infinite horizon setting is given as a function of $h$, $B$, and $T$:

$$\mathrm{MSE}_\infty(h, B, T) = \mathbb{E}\big[(\hat{V}_M(h) - V_\infty)^2\big]. \tag{6}$$

# 3 Characterizing the Mean-Squared Error (MSE)

In the following our aim is to characterize the MSE of the Monte-Carlo estimator as a function of the step-size $h$ and data budget $B$ (and estimation horizon $T$ in the infinite horizon setting). Our results uncover a fundamental trade-off for choosing an *optimal* step-size that leads to a minimal MSE.

**One-Dimensional Langevin Process** To simplify the exposition while preserving the main ideas, we will first present the results for the 1-dimensional case. The analysis for the vector case exhibits the same quantitative behavior but is significantly more involved. To distinguish the 1-dimensional from the $n$-dimensional setting described in Equation (3), we use lower-case symbols. Let $x(t) \in \mathbb{R}$ be the scalar state variable that evolves according to the following Langevin equation:

$$\mathrm{d}x(t) = ax(t)\,\mathrm{d}t + \sigma\,\mathrm{d}w(t). \tag{7}$$

Here, $a \in \mathbb{R}$ is the drift coefficient and $w(t)$ is a Wiener process with scale parameter $\sigma > 0$. We assume that $a \leq 0$, i.e. the system is stable (or marginally stable).

The realized sample path in episode $i = 1, \ldots, M$ is $x_i(t)$ (with starting state $x(0) = 0$) and $t \in [0, T]$. The expected cost is

$$V_\tau = \mathbb{E}\Big[\int_0^\tau \gamma^t r_i(t)\,\mathrm{d}t\Big] = \int_0^\tau \gamma^t q\mathbb{E}\big[x_i^2(t)\big]\,\mathrm{d}t, \tag{8}$$

where $r_i(t) = qx_i^2(t)$ is the quadratic cost function for a fixed $q > 0$. The Riemann sum that approximates the cost realized in episode $i \in [M]$ becomes $\hat{J}_i(h) = \sum_{k=0}^{N-1} hqx_i^2(kh)$. Given data from $M$ episodes, the Monte-Carlo estimator is $\hat{V}_M(h) = \frac{1}{M}\sum_{i=1}^M \hat{J}_i(h)$. Since the square of the cost parameter $q^2$ factors out of the mean-squared error, we set $q = 1$ in what follows.

## 3.1 Finite-Horizon Setting

Recall that in the finite-horizon setting we set the system horizon $\tau$ and estimation horizon $T$ to be the same. This implies that the estimation grid covers the full episode, i.e. $hN = T = \tau$. Perhaps surprisingly, the mean-squared error of the Riemann estimator for the Langevin system (7) can be computed in closed form. The result takes its simplest form in the finite-horizon, undiscounted setting where $\gamma = 1$ and $\tau < \infty$. This result is summarized in the following theorem.

**Theorem 3.1** (Finite-horizon, undiscounted MSE). *In the finite-horizon, undiscounted setting, the mean-squared error of the Monte-Carlo estimator is*

$$\mathrm{MSE}_T(h, B) = E_1(h, T, a) + \frac{E_2(h, T, a)}{B}, \qquad where$$

$$E_1(h, T, a) = \frac{\sigma^4 \left(-2ah + e^{2ah} - 1\right)^2 \left(e^{2aT} - 1\right)^2}{16a^4 \left(e^{2ah} - 1\right)^2},$$

$$E_2(h, T, a) = \frac{\sigma^4 T \left[h \left(e^{2aT} - 1\right)\left(4e^{2ah} + e^{2aT} + 1\right) - \left(e^{2ah} - 1\right)\left(e^{2ah} + 4e^{2aT} + 1\right) T\right]}{2a^2 \left(e^{2ah} - 1\right)^2}.$$

While perhaps daunting at first sight, the result *exactly* characterizes the error surface as a function of the step-size $h$ and the budget $B$ for any given Langevin system. The proof involves computing the closed-form expressions for the second and fourth moments of the random trajectories $x_i(t)$ and is provided in Appendices A and B.1.

In the case of marginal stability ($a = 0$), a simpler form of the MSE emerges that is easier to interpret. Taking the limit $a \to 0$ of the previous expression yields the following result (refer to the discussion and proof in Appendix B.1):

**Corollary 3.2** (MSE for marginally stable system). *Assume a marginally stable system, $a = 0$. Then the mean-squared error of the Monte-Carlo estimator is*

$$\mathrm{MSE}_T(h, B) = \frac{\sigma^4 T^2}{4} \cdot h^2 + \frac{\sigma^4 T^5}{3} \cdot \frac{1}{hB} + \frac{\sigma^4 T^2(-2T^2 + 2hT - h^2)}{3B}.$$

The first part of the expression can be understood as a Riemann sum *approximation error* controlled by the $h^2$ term. The second part corresponds to the *variance term* that decreases with the number of episodes as $\frac{1}{M} = \frac{T}{Bh}$. The remaining terms are of lower order terms for small $h$ and large $B$. For a fixed data budget $B$, the step-size $h$ can be chosen to balance these two terms:

$$h^*(B) := \arg\min_{h>0} \text{MSE}_T(h, B) \approx T \left( \frac{2}{3B} \right)^{1/3}, \tag{9}$$

where the approximation omits higher order terms in $1/B$. From this, we can compute the optimal number of episodes $M^* \approx \frac{Bh^*}{T} = \left( \frac{2}{3} \right)^{1/3} B^{2/3}$. We remark that under the assumption $B \gg 1$, we also obtain that $M^* \gg 1$. This is in agreement with the implicit requirement that $h$ is big enough to consider at least one whole trajectory, i.e. $h > T/B$.

Consequently, the mean-squared error for the optimal choice of $h$ (up to lower order terms in $1/B$):

$$\text{MSE}_T(h^*, B) \approx 3 \, (3/2)^{1/3} \, \sigma^4 T^4 B^{-2/3}.$$

In other words, the optimal error rate as a function of the data budget is $\mathcal{O}(B^{-2/3})$. We can further obtain a similar form for $h^*$ for the general case where $a \leq 0$.

**Corollary 3.3** (Approximate MSE). *The* MSE *is*

$$\text{MSE}_T(h, B) = c_1(\sigma, a, T)h^2 + \frac{c_2(\sigma, a, T)}{hB} + \mathcal{O}(\tfrac{1}{B} + h^3)$$

*for $h \to 0$ and $B \to \infty$, with system-dependent constants*

$$c_1(\sigma, a, T) = \sigma^4 \frac{(e^{2aT} - 1)^2}{16a^2}, \quad c_2(\sigma, a, T) = -\sigma^4 \frac{T \left( 4aT - e^{4aT} + e^{2aT}(8aT - 4) + 5 \right)}{8a^4}.$$

*Moreover, for any $h > 0$ and $B > 0$,*

$$c_1 h^2 + \frac{c_2}{hB} \lesssim \text{MSE}_T(h, B) \lesssim 4c_1 h^2 + \frac{2c_2}{hB}$$

*with $c_1 = c_1(\sigma, a, T)$ and $c_2 = c_2(\sigma, a, T)$ and the inequalities holds true up to a finite, lower-order polynomial expressions $\frac{\sigma^4 h}{B} poly(h, a, T)$, given in Appendix B.2.*

For the proof please see Appendix B.2. From the corollary, we can derive an optimal step-size, up to lower order terms in $1/B$:

$$h^*(B) \approx \left( -\frac{T\left( 4aT - e^{4aT} + e^{2aT}(8aT - 4) + 5 \right)}{a^2 (e^{2aT} - 1)^2} \right)^{1/3} B^{-1/3}. \tag{10}$$

Note that the same $h^*(B)$ also minimizes the upper bound of the MSE up to a constant factor. The scaling $\text{MSE}_T(h^*, B) \leq \mathcal{O}(B^{-2/3})$ cannot be improved given the lower bound on the MSE. The derivation is provided in Appendix B.3 where we also include a more precise expression of $h^*$.

**Discounted Cost**   Adding discounting ($\gamma < 1$) in the finite-horizon setting does not fundamentally change the results; however, it makes the derivation more involved (Appendix B.4).

**Vector Case**   Addressing the general case ($n$-dimensional Langevin systems with $n > 1$) for a stable matrix $A$ requires forgoing the *exact* form of the MSE. We derive tight bounds on the MSE, both of which are convex functions of $h$, thereby narrowing down its behaviour with respect to the step size. The results are presented in Appendix C.3. Although the convex behaviour is proven only for Langevin systems, our experimental results in Section 4 exhibit a similar trade-off for general nonlinear stochastic systems.

Under the additional assumption that the matrix $A$ is also diagonalisable, we are again able to exactly characterise the MSE with closed-form computations. Diagonalisability is a mild assumption since it can be achieved under a controllable system. Indeed, controllability allows for the free adjustment of the eigenvalues of the closed-loop matrix $A$ through the choice of the controller $K$. The eigenvalues can effectively be chosen to be distinct from each other to ensure a diagonalisable $A$. While the explicit form of the MSE is computable, its lengthy formula is not easily interpretable and is thus deferred to Appendix C. The following theorem summarizes the result as a Taylor expansion for small $h$ and large $B$.

**Theorem 3.4** (Mean-squared error - vector case). *Assume $A$ is diagonalisable, with eigenvalues $\Lambda = \{\lambda_1, \ldots, \lambda_n\}$. The mean-squared error of the Monte-Carlo estimator in the finite-horizon, undiscounted setting, is*

$$\text{MSE}_T(h, B) = E_1(h, T, \Lambda) + \frac{E_2(h, T, \Lambda)}{B}, \qquad where$$

$$E_1(h, T, \Lambda) = \left(\overline{C}_1 + C_1(\Lambda)\,\mathcal{O}(T)\right)\sigma^4 T^2 h^2 + \mathcal{O}(h^3)$$

$$\frac{E_2(h, T, \Lambda)}{B} = \left(\overline{C}_2 + C_2(\Lambda)\,\mathcal{O}(T)\right)\sigma^4 \frac{T^5}{hB} + \mathcal{O}(1/B).$$

The proof, including the exact derivation of the constants $\overline{C}_1$, $C_1(\Lambda)$, $\overline{C}_2$, $C_2(\Lambda)$, can be found in Appendix C.1. Note that the terms composing the MSE closely resemble those obtained in the scalar analysis. In fact, when comparing them with the expressions in Equation (22) and Equation (23) (in Appendix B.3), the expression has the same order for $h$, $B$ and $T$. The only difference is that in the vector case, cumbersome eigenvalue-dependent constants are involved, whereas in the scalar case, the result can more easily be expressed in terms of the system parameter $a$.

Since the optimal choice for $h$ is determined by balancing the trade-off between the two terms above, $E_1$ for the approximation error and $E_2$ for the variance, its expression is analogous to the scalar case, as shown by the following corollary.

**Corollary 3.5** (Optimal step size - vector case). *Under the assumption that $B \gg 1$, the optimal step-size for the vector case is given by*

$$h^*(B) = \left(\frac{\overline{C}_1 + C_1(\Lambda)\mathcal{O}(T)}{\overline{C}_2(\Lambda) + C_2(\Lambda)\mathcal{O}(T)}\right)^{1/3} T B^{-1/3} + o\left(B^{-1/3}\right).$$

The constants in Corollary 3.5 are the same as in Theorem 3.4.

## 3.2 Infinite-Horizon Setting

The main characteristic of the finite-horizon setting is the trade-off between approximation and estimation error. Recall that in the infinite-horizon setting ($\tau = \infty$), the estimation horizon $T < \infty$ becomes a free variable that is chosen by the experimenter to define the measurement range $[0, T]$. Consequently the mean-squared error of the Monte-Carlo estimator suffers an additional *truncation error* from using a finite Riemann sum with $N = T/h$ terms as an approximation to the infinite integral that defines the cost $V_\infty$. More precisely, we decompose the expected cost $V_\infty = V_T + V_{T,\infty}$, where $V_T = \int_0^T \gamma^t \mathbb{E}[x^2(t)]dt$ as before, and

$$V_{T,\infty} = \int_T^\infty \gamma^t \mathbb{E}\left[x^2(t)\right]\,\mathrm{d}t = \frac{\sigma^2 \gamma^T}{2a}\left(\frac{1}{\log(\gamma)} - \frac{e^{2aT}}{\log(\gamma) + 2a}\right). \tag{11}$$

It is a direct calculation based on Lemma A.1 in Appendix. Thus the mean-squared error becomes

$$\text{MSE}_\infty(h, B, T) = \mathbb{E}\left[(\hat{V}_M(h) - V)^2\right] = \text{MSE}_T(h, B) - 2\mathbb{E}\left[\hat{V}_M(h) - V_T\right]V_{T,\infty} + V_{T,\infty}^2, \tag{12}$$

where $\text{MSE}_T(h, B) = \mathbb{E}\left[(\hat{V}_M(h) - V_T)^2\right]$ is the mean-squared error of discounted finite-horizon setting. Note that the term $V_{T,\infty}^2$ is neither controlled by a small step-size $h$ nor by a large data budget $B$, hence results in the truncation error from finite estimation. Fortunately, geometric discounting ensures that $V_{T,\infty}^2 = \mathcal{O}(\gamma^{2T})$, which is not unexpected given that the term constitutes the tail of the geometric integral. In particular, setting $T = c \cdot \log(B)/\log(1/\gamma)$ for large enough $c > 1$ ensures the truncation error is below the estimation variance. We summarize the result in the next theorem.

**Theorem 3.6** (Infinite-horizon, discounted MSE). *In the infinite-horizon, discounted setting, the mean-squared error of the Monte-Carlo estimator is*

$$\text{MSE}_\infty(h, B, T) = \sigma^4\, T\, C(a, \gamma) \cdot \frac{1}{hB} + \frac{\sigma^4}{144} \cdot h^4 + \mathcal{O}(h^5) + \mathcal{O}(B^{-1}), \tag{13}$$

*where we let $C(a, \gamma) = \frac{1}{\log(\gamma)(a + \log(\gamma))(2a + \log(\gamma))^2}$ and assume that $\gamma^T = o(h^4)$.*

The proof is provided in Appendix B.5. It follows that the optimal choice for the step-size is $h^*(B, T) \approx (36\, T\, C(a, \gamma)/B)^{1/5}$. The minimal mean-squared error is $\text{MSE}_\infty(h^*, T, B) \leq \mathcal{O}\big((T\, C(a, \gamma)/B)^{4/5} + \gamma^{2T}\big)$. Lastly, we remark that if $\gamma^T$ is treated as a constant, the cross term $\mathbb{E}\big[\hat{V}_M(h) - V_T\big]V_{T,\infty}$ in Equation (12) introduces a dependence of order $\mathcal{O}(h\gamma^{2T})$ to the mean-squared error. In this case, the overall trade-off becomes $\text{MSE}_\infty(h, B, T) \approx \mathcal{O}\big(1/(hB) + \gamma^{2T}(1 + h)\big)$, and the optimal step-size is $h^* \approx B^{-1/2}$.

**Vector Case**    Similar to the finite-horizon setting, we establish tight bounds for the MSE in the general case involving a stable matrix $A$. The detailed results are presented in Appendix C.3. As before, the MSE for the vector case can be computed in closed-form assuming that $A$ is both diagonalisable and stable. The result reflects the same behaviour as in the scalar case. Conveniently, the MSE in Theorem 3.6 has been expressed with sharp terms in $h$ and $B$, while confining the dependence on the system parameter $a$ within the constant $C$, and the impact of higher-order terms in $T$ within $V_{T,\infty}$. This allows us to state the vector case result in a similar form, where the constant will now depend on the eigenvalues of the matrix $A$, as well as the discount factor $\gamma$. These are provided in full detail in Appendix C.2.

**Corollary 3.7.** *For $A$ diagonalisable, with eigenvalues given by $\Lambda$, the mean-squared error of the Monte-Carlo estimator in the infinite-horizon, discounted setting is*

$$\text{MSE}_\infty(h, B, T) = C_3 \sigma^4 \frac{T}{hB} + \frac{\sigma^4}{144} h^4 + \mathcal{O}\left(h^5 + \tfrac{1}{B}\right)$$

*with a constant $C_3 = C_3(\Lambda, \gamma)$ and under the assumption that $\gamma^T = o\left(h^4\right)$.*

The terms in Corollary 3.7 correspond to estimation error, approximation error and truncation error, mirroring the scalar scenario. The optimal step-size exhibits the same dependencies on $T$ and $B$ as in the scalar case, albeit with a different constant dependent on the eigenvalues.

## 4    From Linear to Non-Linear Systems: A Numerical Study

The trade-off identified in our analysis suggests that there exists an optimal choice for temporal resolution in policy evaluation. Our next goal is to verify the trade-off in several simulated dynamical systems. While our analysis assumes a linear transition and quadratic cost, we empirically demonstrate that such a trade-off also exists in nonlinear systems. For our experimental setup, we choose simple linear quadratic systems mirroring the setting of Section 2, as well as several standard benchmarks from Gym [Brockman et al., 2016] and MuJoCo [Todorov et al., 2012]. Our findings confirm the theoretical results and highlight the importance of choosing an appropriate step-size for policy evaluation.

### 4.1    Linear Quadratic Systems

We first run numerical experiments on the Langevin dynamical systems to examine the behaviour of the trade-off identified in our analysis. The results are shown in Fig. 1. For these experiments, we fix the noise $\sigma^2 = 1$ and the cost $Q = I$. The lines in the plot represent the sample mean $(\hat{V}_M(h) - V)^2$ and the shading represent the standard error of our sample means, computed over $50$ independent runs. The plots in Fig. 1 exhibit a clear, U-shaped trade-off, as predicted by our theoretical results.

Fig. 1(a) shows the MSE in a one-dimensional system with $T = 8$ and $a = -1$. The ground truth $V$ is calculated analytically by using Eq. 21 in the Appendix. The figure illustrates how the error changes as we vary the data budget, $B = \{2^{12}, 2^{13}, 2^{14}, 2^{15}, 2^{16}\}$, and also illustrates the improvement that can be obtained by increasing the budget. As we increase $B$, both the error and the optimal step size $h^*$, decrease. This result strongly aligns with the analysis shown in Theorem 3.1 and Corollary 3.3. The same analysis is performed with respect to the parameter $a$, while fixing the data budget, in Fig. 1(b). By increasing the absolute value of the drift coefficient, the diffusion has a smaller impact, thus trajectories have less variability. This leads to a smaller $h^*$ for the optimal point of the trade-off.

Fig. 1(c) and 1(d) present the experimental results for both undiscounted finite horizon and discounted infinite horizon multi-dimensional systems. For the finite-horizon setting, $V$ is computed by numerically solving the Riccati Differential Equation; while in infinite-horizon, it is calculated through Lyapunov equation using a standard solver. In our multi-dimensional experiments, we set the

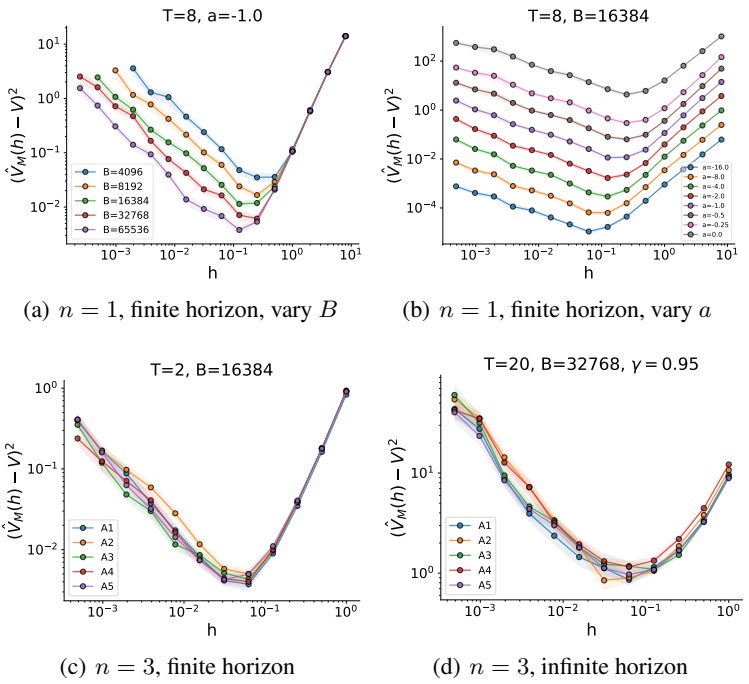

(a) $n = 1$, finite horizon, vary $B$       (b) $n = 1$, finite horizon, vary $a$

(c) $n = 3$, finite horizon           (d) $n = 3$, infinite horizon

Figure 1: Mean-squared error trade-off in linear quadratic systems of different dimension $n$. The first two plots show the dependence of the optimal step-size on the data budget $B$ and drift coefficient $a$, respectively. A{1,2,3,4,5} in the last two plots are random matrices and the two sets are not equal.

dimension $n = 3$. We fix all parameters and run our experiments for 5 randomly sampled $3 \times 3$ dense, stable matrices in each setting. More details on the matrix structure can be found in Appendix D.1. Results in both plots suggest that the impact of the eigenvalues of $A$ on $h^*$ is mild and that the eigenvalue-dependent constant terms in Corollary 3.5 only marginally affect the optimal step-size $h^*$, similar to the trend observed in the scalar case with parameter $a$. In the infinite horizon system, the horizon needs to be large enough to manage truncation error while simultaneously being small enough to collect multiple trajectories. We choose $\gamma$ large enough such that a good estimate of $V$ can be obtained, and set $T = 1/(1 - \gamma)$, which is referred to as the effective horizon in the RL literature.

## 4.2 Nonlinear Systems

Many nonlinear behaviors can be approximated by a high-dimensional linear system, which would be bounded by our theoretical results on the general case of n-dimensional systems, hinting that similar trade-offs could characterize nonlinear systems as well. We empirically show that the trade-off identified in linear quadratic systems carries over to nonlinear systems, with more complex cost functions. We demonstrate it in several simulated nonlinear systems from OpenAI Gym [Brockman et al., 2016], including Pendulum, BipedalWalker and six MuJoCo [Todorov et al., 2012] environments: InvertedDoublePendulum, Pusher, Swimmer, Hopper, HalfCheetah and Ant. We note that the original environments all have a fixed temporal discretization $\delta t$, pre-chosen by the designer. To measure the effect of $h$, we first modify all environments to run at a small discretization $\delta t = 0.001$ as the proxy to the underlying continuous-time systems. We train a nonlinear policy parameterized by a neural network for each system, by the algorithm DAU [Tallec et al., 2019]. This policy is used to gather episode data from the continuous-time system proxy at intervals of $\delta t = 0.001$ which are then down-sampled for different $h$ based on the ratio of $h$ and $\delta t$. The policy is stable in the sense that it produces reasonable behavior (e.g., pendulum stays mostly upright; Ant walking forward; etc.) and not cause early termination of episodes (e.g., BipedalWalker does not fall), in the continuous-time system proxy. The results of the MSE of Monte-Carlo policy evaluation are shown in Fig. 2. Similar to the linear systems case, we vary the data budget $B$ and see how the MSE changes with the step-size $h$. The MSE shows a clear minimum for choosing the optimal step-size $h^*$, which generally decreases as the data budget increases. We slightly abuse notations by using $V, \hat{V}$ to refer to the true and estimated sum of rewards instead of the cost. The true value of $V$ is approximated by averaging the sum of rewards observed

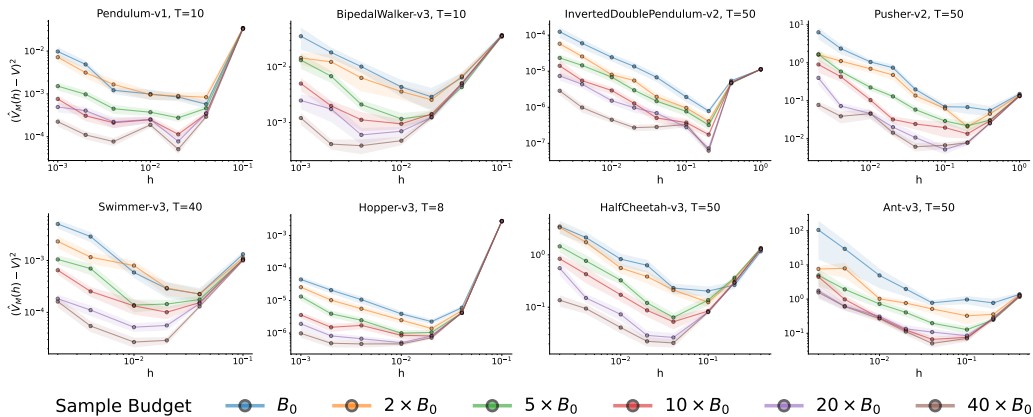

Figure 2: MSE of Monte-Carlo policy evaluation in nonlinear systems. The line and shaded region denote the sample mean and its standard error of $(\hat{V}_M(h) - V)^2$, from 30 random runs. $T$ is the horizon in physical time (seconds). $B_0$ denotes the environment-dependent base sample budget, chosen such that it gives a full episode for the smallest $h$ (see Appendix D.4). The optimal step-size generally decreases as the data budget increases (with 'InvertedDoublePendulum-v2' being the only exception).

at $\delta t = 0.001$ from $150k$ episodes. These environments fall under the finite horizon undiscounted setting. The system (and estimation) horizon $T$ of our experiments is chosen to be the physical time of $1k$ steps under the default $\delta t$ in the original environments (with the exception of 200 steps for Pendulum and 500 steps for BipedalWalker). Please refer to Appendix D.4 for more implementation details, including the setup of $B$, $T$, $\delta t$, $h$, training, and the compute resources. These systems are stochastic in the starting state, while having deterministic dynamics. Despite the different settings from our analysis, a clear trade-off is evident in all systems. This suggests that our findings may have broader applicability than the specific conditions under which our theoretical analysis was established.

### 4.3 Guidelines for Setting Step-Size $h$

The precise characterization of the MSE in Section 3 can be exploited to set the step-size close to the optimal value without any prior knowledge of the system, provided experiments on a smaller data budget are performed beforehand. Although the optimal step-size $h^*$ clearly depends on all quantities characterizing the dynamics and the policy, the technical analysis of the MSE accurately quantifies how $h^*$ scales with respect to the data budget $B$. Specifically, $h^*(B) \approx c_F B^{-1/3}$ for finite horizon and $h^*(B) \approx c_I B^{-1/5}$ for infinite horizon, where $c_F$ and $c_I$ hide the dependencies on the system parameters, exposing only the order in $B$. This allows us to extrapolate the optimal step-size for the given data budget, using the constant estimated with a smaller one. By evaluating through numerical experiments the performances of different step-sizes on the reduced data budget, it is possible to identify an approximate $h^*$, and subsequently determine $c_F$ or $c_I$, which then gives the whole range of optimal values of $h$ with respect to $B$ through the aforementioned relation. Note that this approach does not require prior knowledge of the dynamics, yet it provides a systematic way for setting the step size $h$ for any given scenario.

Figure 3 plots the empirical $h^*$ over $B$ for all nonlinear environments, and fitted lines based on the relation from the analysis $h^* = c_F B^{-1/3}$, where the constant $c_F$ varies with the environment. The plot shows that the scaling of $h^*$ w.r.t. $B$ predicted by our analysis is overall a good approximation of the trend observed in the experiments with nonlinear systems, except for negative cases like the Inverted-DoublePendulum. This suggests that setting the step-size according to the analysis can yield a value close to optimality.

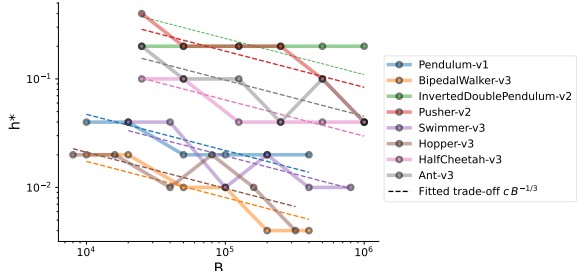

Figure 3: Empirical $h^*$ in nonlinear experiments (solid) compared with analysis in Corollary 3.5 (dashed): $h^* = c_F B^{-1/3}$, $c_F$ is estimated from data by least squares.

# 5 Conclusions

We provide a precise characterization of the approximation, estimation and truncation errors incurred by Monte-Carlo policy evaluation in continuous-time linear stochastic dynamical systems with quadratic cost. This analysis reveals a fundamental bias-variance trade-off, modulated by the level of temporal discretization $h$. Moreover, we confirm in numerical simulations that the analysis accurately captures the trade-off in a precise, quantitative manner. We also demonstrate that the trade-off carries over to non-linear environments such as the popular MuJoCo physics simulation. These findings show that managing the temporal discretization level $h$ can greatly improve the quality of Monte-Carlo policy evaluation under a fixed data budget $B$. These results have direct implications for practice, as it remains common to adopt a pre-set step-size regardless of the data resources anticipated, which we have seen is a highly sub-optimal approach for a given budget.

The present work serves as a first step toward understanding the effects of temporal resolution on RL. There are several limitations that we would like to address in future work. Our analysis is restricted to Monte-Carlo estimation, while there are more advanced techniques such as temporal difference learning and direct system identification, which may exhibit different behaviours. Also, our focus is policy evaluation. It is worth studying policy optimization to understand the full control setting, which might require relaxing the stability assumption on the closed-loop system since the controllers being iterated might not always be stable. Additionally, it would be interesting to explore non-uniform discretization schemes, such as through an adaptive sampling scheme. From the system's perspective, we have currently analyzed stochastic linear quadratic systems with additive Gaussian noise and noiseless observations. It remains to be determined if the exact characterization is still attainable with other types of noise, in the partially observable case, or with noisy observations. Finally, extending the analysis to non-linear systems would be valuable.

## Acknowledgments and Disclosure of Funding

The authors would like to thank Csaba Szepesvari for the helpful discussions. Zichen Zhang gratefully acknowledges the financial support by an NSERC CGSD scholarship and an Alberta Innovates PhD scholarship during this project. He is thankful for the compute resources generously provided by Digital Research Alliance of Canada (and formerly Compute Canada), which is sponsored through the accounts of Martin Jagersand and Dale Schuurmans. Dale Schuurmans gratefully acknowledges funding from the Canada CIFAR AI Chairs Program, Amii and NSERC. Johannes Kirschner gratefully acknowledges funding from the SNSF Early Postdoc.Mobility fellowship P2EZP2_199781.

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

# Appendix

## A   Moment Calculations

Recall that the solution of the SDE in Equation (7), with $x(0) = 0$, takes the following form:

$$x(t) = \sigma \int_0^t e^{a(t-s)} \, \mathrm{d}w(s).$$

(14)

An integral part of finding the mean-squared error of the Monte-Carlo estimator is the computation of the moments $\mathbb{E}\left[x(t)^2\right], \mathbb{E}\left[x(t)^4\right]$ and $\mathbb{E}\left[x(s)^2 x(t)^2\right]$ when $s \leq t$.

**Lemma A.1.** *Let $x(t)$ be the solution of Equation (7). The second moment of the state variable is*

$$\mathbb{E}\left[x^2(t)\right] = \frac{\sigma^2}{2a}\left(e^{2at} - 1\right).$$

(15)

*For the fourth moment, we get:*

$$\mathbb{E}\left[x(t)^4\right] = \frac{3\sigma^4}{4a^2}\left(e^{2at} - 1\right)^2$$

(16)

*Assuming that $s \leq t$, we further get:*

$$\mathbb{E}\left[x^2(s)x^2(t)\right] = \frac{\sigma^4}{4a^2}(e^{2as} - 1)e^{2at}\left\{(e^{-2as} - e^{-2at}) + 3(1 - e^{-2as})\right\}.$$

(17)

*Proof.*

**(1)**   We start with the second moment $\mathbb{E}\left[x(t)^2\right]$.

$$\mathbb{E}\left[x(t)^2\right] = \sigma^2 e^{2at}\mathbb{E}\left[\left(\int_0^t e^{-as}dw(s)\right)^2\right] = \sigma^2 e^{2at}\int_0^t e^{-2as}ds = \frac{\sigma^2}{2a}(e^{2at} - 1)$$

The calculation makes use of the Itô isometry, which can be stated as:

$$\mathbb{E}\left[\left(\int_0^t z(s) \, \mathrm{d}w(s)\right)^2\right] = \mathbb{E}\left[\int_0^t z(s)^2 \, \mathrm{d}s\right],$$

(18)

for any stochastic process $z(\cdot)$ adapted to the filtration induced by the Wiener process $w(\cdot)$.

**(2)**   Next we compute $\mathbb{E}\left[x(t)^4\right]$ through Itô's formula. Define $y(t) := \int_0^t e^{-au} \, \mathrm{d}w(u)$, so that $\mathrm{d}y(t) = e^{-at} \, \mathrm{d}w(t)$. Thus,

$$\mathrm{d}f(y(t)) = f'(y(t)) \, \mathrm{d}y(t) + \frac{1}{2}f''(y(t)) \, (\mathrm{d}y(t))^2$$

$$= f'(y(t)) e^{-at} \, \mathrm{d}w(t) + \frac{1}{2}f''(y(t)) e^{-2at} \, \mathrm{d}t,$$

for any $f(\cdot)$. By choosing $f(y) = y^4$:

$$f'(y) = 4y^3 \quad \text{and} \quad f''(y) = 12y^2.$$

Therefore, by integration and taking the expectation:

$$\mathbb{E}\left[f\left(y\left(t\right)\right)\right] = \mathbb{E}\left[\int_0^t f'\left(y\left(u\right)\right)e^{-au}\,dw\left(u\right)\right] + \frac{1}{2}\mathbb{E}\left[\int_0^t f''\left(y\left(u\right)\right)e^{-2au}\,du\right]$$

$$= \underbrace{\mathbb{E}\left[\int_0^t 4\left(\int_0^u e^{-av}\,dw\left(v\right)\right)^3 e^{-au}\,dw\left(u\right)\right] + \frac{1}{2}\mathbb{E}\left[\int_0^t 12\left(\int_0^u e^{-av}\,dw\left(v\right)\right)^2 e^{-2au}\,du\right]}_{=0}$$

$$= 6\mathbb{E}\left[\int_0^t\left(\int_0^u e^{-av}e^{-au}\,dw\left(v\right)\right)^2\,du\right]$$

$$= 6\int_0^t \mathbb{E}\left[\left(\int_0^u e^{-av}e^{-au}\,dw\left(v\right)\right)^2\right]\,du \qquad \text{(Itô isometry)}$$

$$= 6\int_0^t\int_0^u e^{-2av}e^{-2au}\,dv\,du$$

$$= \int_0^t e^{-2au}\frac{1}{2a}\left(1 - e^{-2au}\right)\,du$$

$$= \frac{3}{4a^2}\left(1 - e^{-2at}\right)^2$$

From Equation (14) it holds $x\left(t\right) = \sigma e^{at}y\left(t\right)$ so that the second part of the lemma follows.

**(3)** Lastly, we compute $\mathbb{E}\left[x(s)^2 x(t)^2\right]$ for $s \leq t$.

$$\mathbb{E}\left[x^2\left(s\right)x^2\left(t\right)\right] = \sigma^4 e^{2a(s+t)}\mathbb{E}\left[\left(\int_0^s e^{-au}\,dw\left(u\right)\right)^2\left(\int_0^t e^{-au}\,dw\left(u\right)\right)^2\right]$$

$$= \sigma^4 e^{2a(s+t)}\mathbb{E}\left[\left(\int_0^s e^{-au}\,dw\left(u\right)\right)^2\left(\int_0^s e^{-au}\,dw\left(u\right) + \int_s^t e^{-au}\,dw\left(u\right)\right)^2\right]$$

$$= \sigma^4 e^{2a(s+t)}\left\{\underbrace{\mathbb{E}\left[\left(\int_0^s e^{-au}\,dw\left(u\right)\right)^4\right]}_{(i)} + \underbrace{\mathbb{E}\left[\left(\int_0^s e^{-au}\,dw\left(u\right)\right)^2\right]\mathbb{E}\left[\left(\int_s^t e^{-au}\,dw\left(u\right)\right)^2\right]}_{(ii)}\right\}$$

Note that we computed (i) before. For (ii) it holds:

$$\mathbb{E}\left[\left(\int_0^s e^{-au}\,dw\left(u\right)\right)^2\right] = \int_0^s e^{-2au}\,du$$

$$= \frac{1}{2a}(1 - e^{-2as})$$

and

$$\mathbb{E}\left[\left(\int_s^t e^{-au}\,dw\left(u\right)\right)^2\right] = \int_s^t e^{-2au}\,du$$

$$= \frac{1}{2a}(e^{-2as} - e^{-2at})$$

Therefore, assuming $s \leq t$, it holds that:

$$\mathbb{E}\left[x^2\left(s\right)x^2\left(t\right)\right] = \sigma^4 e^{2a(s+t)}\left\{\frac{1}{4a^2}\left(1 - e^{-2as}\right)\left(e^{-2as} - e^{-2at}\right) + \frac{3}{4a^2}\left(1 - e^{-2as}\right)^2\right\}$$

$$= \frac{\sigma^4}{4a^2}(e^{2at} - 1)e^{2as}\left\{(e^{-2at} - e^{-2as}) + 3(1 - e^{-2at})\right\}.$$

$\square$

# B  Calculations of the Mean-Squared Error

## B.1  Undiscounted, Finite-Horizon: Proof of Theorem 3.1

*Proof.* We first note that

$$\mathbb{E}[\hat{V}_M(h)] = \frac{h}{M}\sum_{i=1}^{M}\sum_{k=0}^{N-1}\mathbb{E}[x_i^2(kh)] = h\sum_{k=0}^{N-1}\mathbb{E}[x^2(kh)]$$

where we denote $x(t) = x_1(t)$ for simplicity. Next we expand the mean-squared error

$$\mathbb{E}[(\hat{V}_M(h) - V_T)^2] = \mathbb{E}[\hat{V}_M^2(h)] - 2V_T\mathbb{E}[\hat{V}_M(h)] + V_T^2$$

$$= \frac{h^2}{M^2}\mathbb{E}\left[\left(\sum_{i=1}^{M}\sum_{k=0}^{N-1}x_i^2(kh)\right)^2\right] - 2V_T\mathbb{E}[\hat{V}_M(h)] + V_T^2$$

$$= \frac{h^2}{M^2}\sum_{i,j=1}^{M}\sum_{k,l=0}^{N-1}\mathbb{E}[x_i^2(kh)x_j^2(lh)] - 2V_T\mathbb{E}[\hat{V}_M(h)] + V_T^2$$

$$= \frac{h^2}{M}\sum_{k,l=0}^{N-1}\mathbb{E}[x^2(kh)x^2(lh)] + \frac{M^2-M}{M^2}\mathbb{E}[\hat{V}_M(h)]^2 - 2V_T\mathbb{E}[\hat{V}_M(h)] + V_T^2$$

For the last equality, note that $\mathbb{E}[\hat{V}_M(h)]^2 = h^2\sum_{k,l=0}^{N-1}\mathbb{E}[x^2(kh)]\mathbb{E}[x^2(lh)]$. It remains to compute the expressions. By Lemma A.1 we have for the second moment of the state variable:

$$\mathbb{E}[x^2(t)] = \frac{\sigma^2}{2a}\left(e^{2at}-1\right). \tag{19}$$

Assuming that $s \leq t$, from the same lemma we get the following for the fourth moments:

$$\mathbb{E}[x^2(s)x^2(t)] = \frac{\sigma^4}{4a^2}(e^{2as}-1)e^{2at}\left\{(e^{-2as}-e^{-2at}) + 3(1-e^{-2as})\right\}. \tag{20}$$

Note that by symmetry, a similar expression follows for $s \geq t$.

Using these expressions, for the expected cost we get

$$V_T = \int_0^T \mathbb{E}[x^2(t)]dt = \frac{\sigma^2}{2a}\int_0^T\left(e^{2at}-1\right)dt = \frac{\sigma^2}{2a}\left(\frac{e^{2aT}-1}{2a}-T\right) \tag{21}$$

We remark that a similar expression was previously obtained in [Bijl et al., 2016, Theorem 3]. Next, the expected estimated cost is

$$\mathbb{E}[\hat{V}_M(h)] = h\sum_{k=0}^{N-1}\mathbb{E}[x^2(kh)] = \frac{\sigma^2 h}{2a}\sum_{k=0}^{N-1}\left(e^{2akh}-1\right) = \frac{\sigma^2 h}{2a}\left[\frac{1-e^{2aT}}{1-e^{2ah}}-N\right]$$

Lastly, it remains to compute the sum

$$\frac{h^2}{M}\sum_{k,l=0}^{N-1}\mathbb{E}[x^2(kh)x^2(lh)] = \frac{2h^2}{M}\sum_{k<l}^{N-1}\mathbb{E}[x^2(kh)x^2(lh)] + \frac{h^2}{M}\sum_{k=0}^{N-1}\mathbb{E}[x^4(kh)]$$

$$= \frac{\sigma^4 T\left(h^2\left(e^{2aT}-1\right)\left(8e^{2ah}+3e^{2aT}+1\right) + T^2\left(e^{2ah}-1\right)^2 - 2hT\left(e^{2ah}-1\right)\left(e^{2ah}+5e^{2aT}\right)\right)}{4a^2 Bh\left(e^{2ah}-1\right)^2}$$

The last equality is a cumbersome calculation that involves nested geometric sums. We verified the result using symbolic computation. For reference we provide the notebooks containing all calculations in the supplementary material. It remains to collect all terms to get the final result.  □

*Proof of Corollary 3.2.* When $a = 0$, the Langevin equation 7 is reduced to $dx(t) = \sigma\,dw(t)$. The computation of MSE can be performed similarly to that in the proof of Theorem 3.1, by using the following moment results of Wiener process.

$$\mathbb{E}[x^2(t)] = \sigma^2 t,$$

$$\mathbb{E}[x^2(s)x^2(t)] = \sigma^4 s(t+2s) \quad \text{when } s \leq t.$$

It remains to compute the sums, and collect terms, as in the proof of Theorem 3.1.  □

It is also worth pointing out that the result of Corollary 3.2 can also be computed by taking the limit of the MSE in Theorem 3.1 when $a \to 0$. And the resulting MSE from the limit matches the one computed directly as in the proof above. This shows that the MSE in Theorem 3.1 is continuous at $a = 0$.

## B.2 Undiscounted, Finite-Horizon: Approximate MSE

*Proof of Corollary 3.3.* For the asymptotic expansion, we use Theorem 3.1 to compute the leading terms in $h$ of the mean-squared error:

$$E_1(h,T,a) = \frac{\sigma^4(e^{2aT}-1)^2}{16a^2}h^2 + \mathcal{O}(h^3), \tag{22}$$

$$\frac{E_2(h,T,a)}{B} = -\frac{\sigma^4 T\left(4aT - e^{4aT} + e^{2aT}(8aT-4)+5\right)}{8a^4} \cdot \frac{1}{hB} + \frac{\sigma^4 T(1 - e^{4aT} + 4aTe^{2aT})}{4a^3 B}$$

$$- \frac{\sigma^4 Th\left(1 + 4aT + e^{2aT}(8aT+4) - 5e^{4aT}\right)}{24a^2 B} - \frac{\sigma^4 Th^2(e^{4aT}-1)}{12aB} + \mathcal{O}\left(h^3/B\right). \tag{23}$$

Next, we compute explicit upper and lower bounds on the MSE that hold for any $h$ and $B$. Note that for all $x \le 0$,

$$x^2/4 \le \frac{(-x + e^x - 1)^2}{(e^x-1)^2} \le x^2. \tag{24}$$

Hence we can directly bound the $E_1$ term from Theorem 3.1:

$$\frac{\sigma^4 4a^2h^2(e^{2aT}-1)^2}{64a^4} = \frac{\sigma^4 h^2(e^{2aT}-1)^2}{16a^2} \le E_1 \le 4 \cdot \frac{\sigma^4 a^2h^2(e^{2aT}-1)^2}{16a^4}$$

To bound $E_2$ note that for $x \le 0$,

$$\frac{1}{x^2} \le \frac{1}{(1-e^x)^2} \le 1 + \frac{2}{x^2} \tag{25}$$

Abbreviating $E_3 = h\left(e^{2aT}-1\right)\left(4e^{2ah} + e^{2aT}+1\right) - \left(e^{2ah}-1\right)\left(e^{2ah} + 4e^{2aT}+1\right)T$, we have $E_2 = \frac{\sigma^4 TE_3}{2a^2(1-e^{2ah})^2}$, and hence

$$\frac{1}{8a^4h^2} \cdot \sigma^4 T \cdot E_3 \le E_2 \le \left(1 + \frac{1}{8a^4h^2}\right) \cdot E_3 \cdot \sigma^4 T$$

To upper bound $E_2$, we repetitively use that for all $x \le 0$, $1 + x \le e^x \le 1 + x + \frac{x^2}{2}$ and $1 + x + \frac{x^2}{2} + \frac{x^3}{6} \le e^x$.

$$E_3 = h\left(e^{2aT}-1\right)\left(4e^{2ah} + e^{2aT}+1\right) - \left(e^{2ah}-1\right)\left(e^{2ah} + 4e^{2aT}+1\right)T$$

$$= 4he^{2aT}e^{2ah} + he^{4aT} - 4he^{2ah} - h - Te^{4ah} - 4Te^{2ah}e^{2aT} + 4Te^{2aT} + T$$

$$= h(e^{2ah}(4e^{2aT}-4) + e^{4aT}-1) - 4Te^{2aT}e^{2ah} - Te^{4ah} + 4Te^{2aT} + T$$

$$\le h\left((1 + 2ah + 2a^2h^2 + \frac{4}{3}a^3h^3)(4e^{2aT}-4) + e^{4aT}-1\right)$$

$$- 4Te^{2aT}(1 + 2ah + 2a^2h^2 + \frac{4}{3}a^3h^3) - T(1 + 4ah + 8a^2h^2 + \frac{32}{3}a^3h^3) + 4Te^{2aT} + T$$

$$= h(4e^{2aT} - 5 + e^{4aT} - 8aTe^{2aT} - 4Ta) + h^2(2a(4e^{2aT}-4) - 8a^2Te^{2aT} - 8a^2T)$$

$$+ h^3(8a^2(e^{2aT}-1) - \frac{16}{3}a^3Te^{2aT} - \frac{32}{3}a^3) + h^4\frac{16}{3}(e^{2aT}-1)$$

$$\le h \cdot (4e^{2aT} - 5 + e^{4aT} - 8aTe^{2aT} - 4Ta) + \frac{32}{3}h^2a^4T^3 + 16h^3a^4T^2$$

Combining the last two displays yields the claimed upper bound. The lower bound follows along the same lines. Note that the bounds can be refined by including higher-order approximations of $e^x$.

$\square$

## B.3 Undiscounted, Finite-Horizon: Optimal Step Size

Although the exact optimal step size $h^*$ can be obtained from Theorem 3.1, such exact $h^*$ doesn't have an explicit analytic solution in general. Numerically, we can find $h^*$ by searching over step-sizes $h_m = T/m$ for $m = 1, \ldots, B$, provided knowledge of the system parameters $a$ and fixed horizon $T$ or, by finding the root between 0 and 1 of the following equation

$$[5 + 4aT - e^{2aT}(4 + e^{2aT} - 8aT)](9aTh + 3T) + 2a^2Th^2(37 - 5a^{4aT} + 28aT + 56aTe^{2aT}$$
$$- 32e^{2aT}) + a^2h^3[3B(e^{2aT} - 1)^2 + aT(91 - 7e^{4aT} + 60aT + 120aTe^{2aT} - 84e^{2aT})] = 0\,,$$

where the equation is a simplified form of $\frac{\partial}{\partial h}\mathrm{MSE}_T(h, B) = 0$.

From the analysis point of view, a trivial way to see the order of $h^*$ in terms of $B, a, T$ is finding the dominated term by using Taylor's expansion for exponential parts (which is true for any $h$) in Theorem 3.1. Such asymptotic expansion is given in Corollary 3.3. It is immediate that for $h \geq 1$, both Equation (22) and Equation (23) will blow up. Thus, a small $h < 1$ is considered to minimize $E_1(h, T, a) + \frac{E_2(h,T,a)}{B}$. Keeping the first term in both Equation (22) and Equation (23) and solving for the optimal $h^*$ yields the result in Equation (10).

A more precise approximation of $h^*$ than Equation (10) is a minimizer of $E_1(h, T, a) + \frac{E_2(h,T,a)}{B}$ truncated at $\mathcal{O}(h^3)$:

$$h^*(a, T, B) = \frac{D_1}{3D_3} + \left( \frac{D_1^3}{3^3 D_3^3} - \frac{3D_2}{2a^2 D_3} - \sqrt{\frac{9D_2^2}{4a^4 D_3^2} - \frac{D_1^3 D_2}{9a^2 D_3^4}} \right)^{\frac{1}{3}}$$
$$+ \left( \frac{D_1^3}{3^3 D_3^3} - \frac{3D_2}{2a^2 D_3} + \sqrt{\frac{9D_2^2}{4a^4 D_3^2} - \frac{D_1^3 D_2}{9a^2 D_3^4}} \right)^{\frac{1}{3}}\,, \tag{26}$$

where

$$D_1 = T\left(1 + 4aT + e^{2aT}(8aT + 4) - 5e^{4aT}\right)\,,$$
$$D_2 = T\left(4aT - e^{4aT} + e^{2aT}(8aT - 4) + 5\right)\,,$$
$$D_3 = 3B(e^{2aT} - 1)^2 - 4aT(e^{4aT} - 1)\,.$$

We can further express Equation (26) in terms of $B$, as

$$h^*(B) = \left( -\frac{T\left(4aT - e^{4aT} + e^{2aT}(8aT - 4) + 5\right)}{a^2(e^{2aT} - 1)^2} \right)^{1/3} B^{-1/3}$$
$$+ \frac{T\left(1 + 4aT + e^{2aT}(8aT + 4) - 5e^{4aT}\right)}{9(e^{2aT} - 1)^2 B} + \frac{4aT(e^{2aT} + 1)D_2^{1/3}}{9a^{2/3}(e^{2aT} - 1)^{5/3}} B^{-4/3}$$
$$+ \frac{4aT^2(e^{2aT} + 1)D_1}{27(e^{2aT} - 1)^3} B^{-2} + \mathcal{O}(B^{-7/3})\,.$$

where the first term is exactly the result in Equation (10).

## B.4 Finite-Horizon, Discounted

As stated in Section 3.1, adding discounting in the finite-horizon setting makes the mean-squared error more involved. In the regime where $h$ is small and $B$ is large, a Taylor expansion characterizes the error surface as follows:

$$\mathrm{MSE}_T(h, B, \gamma) \approx \frac{\sigma^4 T}{\log(\gamma)(a + \log(\gamma))(2a + \log(\gamma))^2} \cdot \frac{1}{hB} + \frac{\sigma^4 \gamma^{2T}(e^{2aT} - 1)^2}{16a^2} \cdot h^2$$
$$+ \frac{\sigma^4 \gamma^T\left(e^{2aT} - 1\right)\left(\gamma^T\left(e^{2aT}(2a + \log(\gamma)) - \log(\gamma)\right) - 2a\right)}{48a^2} \cdot h^3 + \frac{\sigma^4}{144} \cdot h^4$$
$$\tag{27}$$

The approximation shows only the lowest order terms for $1/(hB)$, $\gamma^T$ and $h$. The derivation is given in Lemma B.1 below. The results shows that main trade-off between $h$ and $B$ persists also for the discounted objective, as long as $\gamma^T$ is treated as a constant relative to $h^2$ and $1/hB$. In the limit where $\gamma^T$ becomes small (e.g. $\gamma^T = o(h^4)$) the nature of the trade-off changes in that the approximation error improves to $\mathcal{O}(h^4)$. This can be understood from the fact that under geometric discounting combined with a decaying process, the sum of $N = T/h$ estimation errors do not suffer a factor $N$, thereby removing a factor of $1/h$ from the (non-squared) approximation error.

**Lemma B.1** (Finite-horizon, discounted). *In the finite-horizon with a discount factor $\gamma \in (0, 1]$ setting, the mean-squared error of the Monte-Carlo estimator is*

$$\text{MSE}_T(h, B, \gamma) = E_1(h, T, a, \gamma) + \frac{E_2(h, T, a, \gamma)}{B},$$

*where*

$$E_1(h, T, a, \gamma) = C_1(T, \gamma, a)\sigma^4 h^2 + C_2(T, \gamma, a)\sigma^4 h^3 + \left( \frac{1}{144} + C_3(T, \gamma, a) \right) \sigma^4 h^4 + \mathcal{O}(h^5),$$

$$E_2(h, T, a, \gamma) = \frac{\sigma^4(T + \gamma^T C_4(T, \gamma, a))}{\log(\gamma)(a + \log(\gamma))(2a + \log(\gamma))^2 h} + \gamma^T \mathcal{O}(1),$$

$$C_1(T, \gamma, a) = \frac{\gamma^{2T} \left( e^{2aT} - 1 \right)^2}{16a^2},$$

$$C_2(T, \gamma, a) = \frac{\gamma^T \left( e^{2aT} - 1 \right) \left( \gamma^T \left( e^{2aT}(2a + \log(\gamma)) - \log(\gamma) \right) - 2a \right)}{48a^2},$$

$$C_3(T, \gamma, a) = \frac{\gamma^T \left[ \gamma^T \left( e^{2aT}(2a + \log(\gamma)) - \log(\gamma) \right)^2 - 4a \left( e^{2aT}(2a + \log(\gamma)) - \log(\gamma) \right) \right]}{576a^2},$$

*$C_4(T, \gamma, a)$ is some finite constant of $(T, \gamma, a)$ that includes $\gamma^T$ as a factor.*

*Proof.* The proof follows the similar computations as those in the previous proof with a new expected cost as follows. In particular, using Lemma A.1, we get

$$V_T = \int_0^T \gamma^t \mathbb{E}[x^2(t)]dt = \frac{\sigma^2}{2a} \left( \frac{\gamma^T e^{2aT} - 1}{\log(\gamma) + 2a} - \frac{\gamma^T - 1}{\log(\gamma)} \right) \tag{28}$$

Furthermore, the expected estimated cost is

$$\mathbb{E}[\hat{V}_M(h)] = \frac{\sigma^2 h}{2a} \sum_{k=0}^{N-1} \gamma^{kh} \left( e^{2akh} - 1 \right) = \frac{\sigma^2 h}{2a} \left( \frac{1 - \gamma^T e^{2aT}}{1 - \gamma^h e^{2ah}} - \frac{1 - \gamma^T}{1 - \gamma^h} \right).$$

Finally, the sum containing the fourth order cross-moments is

$$\frac{h^2}{M} \sum_{k,l=0}^{N-1} \gamma^{kh+lh} \mathbb{E}[x^2(kh)x^2(lh)] = \frac{2h^2}{M} \sum_{k<l}^{N-1} \gamma^{kh+lh} \mathbb{E}[x^2(kh)x^2(lh)] + \frac{h^2}{M} \sum_{k=0}^{N-1} \gamma^{2kh} \mathbb{E}[x^4(kh)].$$

While not impossible to calculate on paper, a written derivation is beyond the scope of this work. Instead, we rely on symbolic computation to obtain the expression and corresponding Taylor approximations. The notebooks containing all derivations are provided in the supplementary material. $\square$

## B.5 Infinite Horizon: Proof of Theorem 3.6

*Proof.* The proof relies on the decomposition provided in Equation (12). It only remains to compute the following cross term.

$$\mathbb{E}\left[ \hat{V}_M(h) - V_T \right] V_{T,\infty}$$

$$= \frac{\sigma^4}{2a} \left( \frac{\gamma^T}{\log(\gamma)} - \frac{\gamma^T e^{2aT}}{\log(\gamma) + 2a} \right) \left[ \frac{h}{2a} \left( \frac{1 - \gamma^T e^{2aT}}{1 - \gamma^h e^{2ah}} - \frac{1 - \gamma^T}{1 - \gamma^h} \right) - \frac{1}{2a} \left( \frac{\gamma^T e^{2aT} - 1}{\log(\gamma) + 2a} - \frac{\gamma^T - 1}{\log(\gamma)} \right) \right]$$

$$= \frac{\sigma^4 \gamma^{2T} \left( e^{2aT} - 1 \right) \left( \log(\gamma) \left( e^{2aT} - 1 \right) - 2a \right)}{8a^2 \log(\gamma) \left( 2a + \log(\gamma) \right)} h +$$

$$\frac{\sigma^4 \gamma^T \left( 2a + \log(\gamma) - e^{2aT} \log(\gamma) \right) \left( 2a \left( \gamma^T e^{2aT} - 1 \right) + \gamma^T \log(\gamma) \left( e^{2aT} - 1 \right) \right)}{48a^2 \log(\gamma) \left( 2a + \log(\gamma) \right)} h^2 + \mathcal{O}(h^3).$$

Thus, the mean-squared error $\mathrm{MSE}_\infty(h, B, T, \gamma) = \mathbb{E}\big[(\hat{V}_M(h) - V_\infty)^2\big]$ is obtained by combining the above computation with Equation (11) and Lemma B.1. $\qquad\square$

It is worth pointing out the trade-off always exists with or without the assumption $\gamma^T = o(h^4)$ from $\mathrm{MSE}_\infty(h, B, T, \gamma)$. For example, if $\gamma^T$ is constant with respect to $h$,

$$
\begin{aligned}
\mathrm{MSE}_\infty(h, B, T, \gamma) = {}& \sigma^4\, T\, \frac{1}{\log(\gamma)(a + \log(\gamma))(2a + \log(\gamma))^2} \cdot \frac{1}{hB} \\
& + \frac{\sigma^4 \gamma^{2T} \left(e^{2aT} - 1\right)\left(\log(\gamma)\left(e^{2aT} - 1\right) - 2a\right)}{8a^2 \log(\gamma)\left(2a + \log(\gamma)\right)} h \\
& + \frac{\sigma^4 \gamma^{2T}[(1 - e^{2aT})\log(\gamma) + 2a]^2}{4a^2[\log(\gamma)(a + \log(\gamma))]^2} + \mathcal{O}(h^2) + \mathcal{O}(B^{-1}).
\end{aligned}
$$

From the above expression, $\mathrm{MSE}_\infty(h, B, T, \gamma)$ contains a constant term in this case. For the other cases of $\gamma^T$, $\mathrm{MSE}_\infty(h, B, T, \gamma)$ can be obtained similarly by combining the cross term in the proof of Theorem 3.6, Equation (11) and Lemma B.1. The case where $\gamma^T = o(h^4)$ is particularly interesting because even as $\gamma^T \to 0$ at such a fast rate, there is still a trade-off that never vanishes.

## C  Vector Case Analysis

### C.1  Finite-Horizon, Undiscounted: Proof of Theorem 3.4

*Proof.* Consider the n-dimensional system that the solution of the trajectory of $X(t)$ is

$$
X(t) = \sigma \int_0^t e^{A(t-s)}\, \mathrm{d}W(t).
$$

Since $A$ is a diagonalizable matrix, we can decompose $A$ as $A = P^{-1}DP$, where $P$ is a invertible matrix (not necessarily to be orthogonal) and $D$ is a diagonal matrix whose diagonal entries $(\lambda_1, \cdots, \lambda_n)$ are corresponding to the eigenvalues of the matrix $A$. Followed by which, we can decompose the matrix exponential of $A$ as:

$$
e^{At} = P^{-1}e^{Dt}P.
$$

Define the "diagonalized" process $\tilde{X}(\cdot)$ as:

$$
\begin{aligned}
PX(t) &= P\sigma \int_0^t e^{A(t-s)}\, \mathrm{d}W(s) \\
&= \sigma P P^{-1} \int_0^t e^{D(t-s)}P\, \mathrm{d}W(s) \\
&= \sigma \int_0^t e^{D(t-s)}\, \mathrm{d}\tilde{W}(s) =: \tilde{X}(t)
\end{aligned}
$$

where $\tilde{W}(s)$ is a Wiener process (with dependent components when $P$ is not orthogonal). This implies that $X(\cdot) = P^{-1}\tilde{X}(\cdot)$.

To see $\tilde{X}_i(t)$ clearly, we denote $P = [p_{ij}]_{i,j=1}^n$, and $\tilde{X}_i(t) = (\phi_1^{(i)}(t), \cdots, \phi_n^{(i)}(t))^\top$, then $\phi_l^{(i)}(t) = \sum_{j=1}^n p_{lj}\sigma \int_0^t e^{\lambda_l(t-s)}\, \mathrm{d}w_j^{(i)}(s)$ for each $l \in \{1, \cdots, n\}$. Particularly, in such an expression, $w_j^{(i)}(s)$ are independent Wiener processes for different $i$ or $j$. Correspondingly, $\tilde{X}(t) = (\phi_1(t), \cdots, \phi_n(t))^\top$, and $\phi_l(t) = \sum_{j=1}^n p_{lj}\sigma \int_0^t e^{\lambda_l(t-s)}\, \mathrm{d}w_j(s)$ for each $l \in \{1, \cdots, n\}$, where $w_j(s)$ are independent Wiener processes for different $j$.

By trace operation, we can rewrite $\hat{V}_M(h)$ as follows:

$$\hat{V}_M(h) = \frac{1}{M}\sum_{i=1}^{M}\sum_{k=0}^{N-1} hX(t_k)^\top Q X(t_k)$$

$$= \mathrm{tr}\left\{\frac{1}{M}\sum_{i=1}^{M}\sum_{k=0}^{N-1} h\tilde{X}(t_k)^\top P^{-\top}QP^{-1}\tilde{X}(t_k)\right\}$$

$$= \mathrm{tr}\left\{P^{-\top}QP^{-1}\hat{\mathcal{V}}_M(h)\right\},$$

where $\hat{\mathcal{V}}_M(h) = \frac{1}{M}\sum_{i=1}^{M}\sum_{k=0}^{N-1} h\tilde{X}(t_k)\tilde{X}(t_k)^\top \in \mathbb{R}^{n\times n}$.

Similarly, $V_T = \mathrm{tr}\left\{P^{-\top}QP^{-1}\mathcal{V}_T\right\}$, where $\mathcal{V}_T = \int_0^T \mathbb{E}[\tilde{X}(t)\tilde{X}(t)^\top]\,\mathrm{d}t$.

Therefore, the $\mathrm{MSE}_T(h,B)$ can be written as

$$\mathrm{MSE}_T(h,B) = \mathbb{E}\left[\left(\hat{V}_M(h) - V_T\right)^2\right] = \mathbb{E}\left[\mathrm{tr}\left\{P^{-\top}QP^{-1}\left(\hat{\mathcal{V}}_M(h) - \mathcal{V}_T\right)\right\}^2\right]. \tag{29}$$

For notional simplicity, we denote matrix $P^{-\top}QP^{-1} =: B = [b_{lj}]_{l,j=1}^n$ and $\hat{\mathcal{V}}_M(h) - \mathcal{V}_T =: C = [c_{lj}]_{l,j=1}^n$.

Noting the fact that

$$\mathrm{MSE}_T(h,B) = \mathbb{E}\left[\left(\sum_{l,j} b_{jl}c_{lj}\right)^2\right] = \sum_{l_1,j_1,l_2,j_2} b_{j_1 l_1}b_{j_2 l_2}\mathbb{E}\left[c_{l_1 j_1}c_{l_2 j_2}\right], \tag{30}$$

it is sufficient to find $\mathrm{MSE}_T$ by only computing $\mathbb{E}\left[c_{l_1 j_1}c_{i_2 j_2}\right]$.

We first introduce the following expectations that are used in the computations. For any $s\leq t$

$$\mathbb{E}\left[\int_0^t e^{\lambda_1(t-u)}\,\mathrm{d}w(u)\int_0^s e^{\lambda_2(s-u)}\,\mathrm{d}w(u)\right] = \frac{e^{\lambda_1 t+\lambda_2 s}}{\lambda_1+\lambda_2}\left(1 - e^{-(\lambda_1+\lambda_2)s}\right), \tag{31}$$

$$\mathbb{E}\left[\int_0^s e^{\lambda_1(s-u)}\,\mathrm{d}w(u)\int_0^s e^{\lambda_2(s-u)}\,\mathrm{d}w(u)\int_0^t e^{\lambda_3(t-u)}\,\mathrm{d}w(u)\int_0^t e^{\lambda_4(t-u)}\,\mathrm{d}w(u)\right]$$

$$= e^{(\lambda_1+\lambda_2)s+(\lambda_3+\lambda_4)t}\left[\frac{1}{(\lambda_1+\lambda_2)(\lambda_3+\lambda_4)}\left(1 - e^{-(\lambda_1+\lambda_2)s}\right)\left(1 - e^{-(\lambda_3+\lambda_4)s}\right)\right.$$

$$+ \frac{1}{(\lambda_1+\lambda_3)(\lambda_2+\lambda_4)}\left(1 - e^{-(\lambda_1+\lambda_3)s}\right)\left(1 - e^{-(\lambda_2+\lambda_4)s}\right)$$

$$+ \frac{1}{(\lambda_1+\lambda_4)(\lambda_2+\lambda_3)}\left(1 - e^{-(\lambda_1+\lambda_4)s}\right)\left(1 - e^{-(\lambda_2+\lambda_3)s}\right)$$

$$\left.+ \frac{1}{(\lambda_1+\lambda_2)(\lambda_3+\lambda_4)}\left(1 - e^{-(\lambda_1+\lambda_2)s}\right)\left(e^{-(\lambda_3+\lambda_4)s} - e^{-(\lambda_3+\lambda_4)t}\right)\right] \tag{32}$$

$$\int_0^T \mathbb{E}\left[\int_0^t e^{\lambda_1(t-u)}\,\mathrm{d}w(u)\int_0^s e^{\lambda_2(-u)}\,\mathrm{d}w(u)\right]\,\mathrm{d}t = \frac{e^{(\lambda_1+\lambda_2)T} - 1 - (\lambda_1+\lambda_2)T}{(\lambda_1+\lambda_2)^2}. \tag{33}$$

By using the definitions of $\hat{\mathcal{V}}_M(h)$ and $\mathcal{V}_T$, it is trivial to see for any $l, j \in \{1, \cdots, n\}$

$$
\begin{aligned}
c_{lj} &= \frac{1}{M} \sum_{i=1}^{M} \sum_{k=0}^{N-1} h\phi_l^{(i)}(kh)\phi_j^{(i)}(kh) - \int_0^T \mathbb{E}[\phi_l(t)\phi_j(t)]\,\mathrm{d}t \\
&= \frac{h\sigma^2}{M} \sum_{i=1}^{M} \sum_{k=0}^{N-1} \left( \sum_{\alpha=1}^{n} p_{l\alpha} \int_0^{kh} e^{\lambda_l(kh-s)}\,\mathrm{d}w_\alpha^{(i)}(s) \right) \left( \sum_{\alpha=1}^{n} p_{j\alpha} \int_0^{kh} e^{\lambda_j(kh-s)}\,\mathrm{d}w_\alpha^{(i)}(s) \right) \\
&\quad - \sigma^2 \int_0^T \mathbb{E}\left[ \left( \sum_{\alpha=1}^{n} p_{l\alpha} \int_0^t e^{\lambda_l(t-s)}\,\mathrm{d}w_\alpha(s) \right) \left( \sum_{\alpha=1}^{n} p_{j\alpha} \int_0^t e^{\lambda_j(t-s)}\,\mathrm{d}w_\alpha(s) \right) \right]\,\mathrm{d}t \\
&= \sum_{\alpha=1}^{n} p_{l\alpha} p_{j\alpha} \left[ \frac{h\sigma^2}{M} \sum_{i=1}^{M} \sum_{k=0}^{N-1} \left( \int_0^{kh} e^{\lambda_l(kh-s)}\,\mathrm{d}w_\alpha^{(i)}(s) \right) \left( \int_0^{kh} e^{\lambda_j(kh-s)}\,\mathrm{d}w_\alpha^{(i)}(s) \right) \right. \\
&\quad \left. - \sigma^2 \int_0^T \mathbb{E}\left[ \left( \int_0^t e^{\lambda_l(t-s)}\,\mathrm{d}w_\alpha(s) \right) \left( \int_0^t e^{\lambda_j(t-s)}\,\mathrm{d}w_\alpha(s) \right) \right]\,\mathrm{d}t \right] + \\
&\quad \sum_{\alpha \neq \beta} p_{l\alpha} p_{j\beta} \left[ \frac{h\sigma^2}{M} \sum_{i=1}^{M} \sum_{k=0}^{N-1} \left( \int_0^{kh} e^{\lambda_l(kh-s)}\,\mathrm{d}w_\alpha^{(i)}(s) \right) \left( \int_0^{kh} e^{\lambda_j(kh-s)}\,\mathrm{d}w_\beta^{(i)}(s) \right) \right],
\end{aligned}
$$

where the last equation is due to the fact that for $\alpha \neq \beta$

$$
\mathbb{E}\left[ \left( \int_0^t e^{\lambda_l(t-s)}\,\mathrm{d}w_\alpha(s) \right) \left( \int_0^t e^{\lambda_j(t-s)}\,\mathrm{d}w_\beta(s) \right) \right] = 0.
$$

Thus, for any $l_1, l_2, j_1, j_2 \in \{1, \cdots, n\}$,

$$
\begin{aligned}
\mathbb{E}\left[ c_{l_1 j_1} c_{l_2, j_2} \right] &= \sum_{\alpha=1}^{n} p_{l_1\alpha} p_{j_1\alpha} p_{l_2\alpha} p_{j_2\alpha} \sigma^4 \mathcal{I}_1 \left( M, h, T, \lambda_{l_1}, \lambda_{j_1}, \lambda_{l_2}, \lambda_{j_2}, \alpha \right) \\
&\quad + \sum_{\alpha \neq \beta}^{n} p_{l_1\alpha} p_{j_1\alpha} p_{l_2\beta} p_{j_2\beta} \sigma^4 \mathcal{I}_2 \left( M, h, T, \lambda_{l_1}, \lambda_{j_1}, \lambda_{l_2}, \lambda_{j_2}, \alpha, \beta \right) \\
&\quad + \sum_{\alpha \neq \beta}^{n} p_{l_1\alpha} p_{j_1\beta} p_{l_2\alpha} p_{j_2\beta} \sigma^4 \mathcal{I}_3 \left( M, h, T, \lambda_{l_1}, \lambda_{j_1}, \lambda_{l_2}, \lambda_{j_2}, \alpha, \beta \right), \quad (34)
\end{aligned}
$$

where

$$
\begin{aligned}
&\mathcal{I}_1 \left( M, h, T, \lambda_{l_1}, \lambda_{j_1}, \lambda_{l_2}, \lambda_{j_2}, \alpha \right) \\
&= \mathbb{E}\left\{ \left[ \frac{h}{M} \sum_{i=1}^{M} \sum_{k=0}^{N-1} \left( \int_0^{kh} e^{\lambda_{l_1}(kh-s)}\,\mathrm{d}w_\alpha^{(i)}(s) \right) \left( \int_0^{kh} e^{\lambda_{j_1}(kh-s)}\,\mathrm{d}w_\alpha^{(i)}(s) \right) \right. \right. \\
&\quad \left. - \int_0^T \mathbb{E}\left[ \left( \int_0^t e^{\lambda_{l_1}(t-s)}\,\mathrm{d}w_\alpha(s) \right) \left( \int_0^t e^{\lambda_{j_1}(t-s)}\,\mathrm{d}w_\alpha(s) \right) \right]\,\mathrm{d}t \right] \times \\
&\quad \left[ \frac{h}{M} \sum_{i=1}^{M} \sum_{k=0}^{N-1} \left( \int_0^{kh} e^{\lambda_{l_2}(kh-s)}\,\mathrm{d}w_\alpha^{(i)}(s) \right) \left( \int_0^{kh} e^{\lambda_{j_2}(kh-s)}\,\mathrm{d}w_\alpha^{(i)}(s) \right) \right. \\
&\quad \left. \left. - \int_0^T \mathbb{E}\left[ \left( \int_0^t e^{\lambda_{l_2}(t-s)}\,\mathrm{d}w_\alpha(s) \right) \left( \int_0^t e^{\lambda_{j_2}(t-s)}\,\mathrm{d}w_\alpha(s) \right) \right]\,\mathrm{d}t \right] \right\},
\end{aligned}
$$

and

$$\mathcal{I}_2\left(M, h, T, \lambda_{l_1}, \lambda_{j_1}, \lambda_{l_2}, \lambda_{j_2}, \alpha, \beta\right)$$

$$= \mathbb{E}\Bigg\{\Bigg[\frac{h}{M}\sum_{i=1}^{M}\sum_{k=0}^{N-1}\left(\int_0^{kh}e^{\lambda_{l_1}(kh-s)}\,\mathrm{d}w_\alpha^{(i)}(s)\right)\left(\int_0^{kh}e^{\lambda_{j_1}(kh-s)}\,\mathrm{d}w_\alpha^{(i)}(s)\right)$$

$$- \int_0^T \mathbb{E}\left[\left(\int_0^t e^{\lambda_{l_1}(t-s)}\,\mathrm{d}w_\alpha(s)\right)\left(\int_0^t e^{\lambda_{j_1}(t-s)}\,\mathrm{d}w_\alpha(s)\right)\right]\,\mathrm{d}t\Bigg] \times$$

$$\Bigg[\frac{h}{M}\sum_{i=1}^{M}\sum_{k=0}^{N-1}\left(\int_0^{kh}e^{\lambda_{l_2}(kh-s)}\,\mathrm{d}w_\beta^{(i)}(s)\right)\left(\int_0^{kh}e^{\lambda_{j_2}(kh-s)}\,\mathrm{d}w_\beta^{(i)}(s)\right)$$

$$- \int_0^T \mathbb{E}\left[\left(\int_0^t e^{\lambda_{l_2}(t-s)}\,\mathrm{d}w_\beta(s)\right)\left(\int_0^t e^{\lambda_{j_2}(t-s)}\,\mathrm{d}w_\beta(s)\right)\right]\,\mathrm{d}t\Bigg]\Bigg\}$$

and

$$\mathcal{I}_3\left(M, h, T, \lambda_{l_1}, \lambda_{j_1}, \lambda_{l_2}, \lambda_{j_2}, \alpha, \beta\right)$$

$$= \mathbb{E}\Bigg\{\Bigg[\frac{h}{M}\sum_{i=1}^{M}\sum_{k=0}^{N-1}\left(\int_0^{kh}e^{\lambda_{l_1}(kh-s)}\,\mathrm{d}w_\alpha^{(i)}(s)\right)\left(\int_0^{kh}e^{\lambda_{j_1}(kh-s)}\,\mathrm{d}w_\alpha^{(i)}(s)\right)\Bigg] \times$$

$$\Bigg[\frac{h}{M}\sum_{i=1}^{M}\sum_{k=0}^{N-1}\left(\int_0^{kh}e^{\lambda_{l_2}(kh-s)}\,\mathrm{d}w_\alpha^{(i)}(s)\right)\left(\int_0^{kh}e^{\lambda_{j_2}(kh-s)}\,\mathrm{d}w_\beta^{(i)}(s)\right)\Bigg]\Bigg\}.$$

Note that $w_\alpha^{(i)}$ and $w_\beta^{(i)}$ are independent for $\alpha \neq \beta$. By using the expectations Equations (31) and (33), we can further obtain $\mathcal{I}_2\left(M, h, T, \lambda_{l_1}, \lambda_{j_1}, \lambda_{l_2}, \lambda_{j_2}, \alpha, \beta\right)$ as

$$\mathcal{I}_2\left(M, h, T, \lambda_{l_1}, \lambda_{j_1}, \lambda_{l_2}, \lambda_{j_2}, \alpha, \beta\right)$$

$$= \left[\frac{h}{(\lambda_{l_1} + \lambda_{j_1})}\left(\frac{1 - e^{(\lambda_{l_1}+\lambda_{j_1})T}}{1 - e^{(\lambda_{l_1}+\lambda_{j_1})h}} - \frac{T}{h}\right) - \frac{1}{(\lambda_{l_1} + \lambda_{j_1})^2}\left(e^{(\lambda_{l_1}+\lambda_{j_1})T} - 1 - (\lambda_{l_1} + \lambda_{j_1})T\right)\right]$$

$$\times \left[\frac{h}{(\lambda_{l_2} + \lambda_{j_2})}\left(\frac{1 - e^{(\lambda_{l_2}+\lambda_{j_2})T}}{1 - e^{(\lambda_{l_2}+\lambda_{j_2})h}} - \frac{T}{h}\right) - \frac{1}{(\lambda_{l_2} + \lambda_{j_2})^2}\left(e^{(\lambda_{l_2}+\lambda_{j_2})T} - 1 - (\lambda_{l_2} + \lambda_{j_2})T\right)\right].$$

In the following computations, we will use $\bar{C}$ and $C(\lambda_{l_1}, \lambda_{j_1}, \lambda_{l_2}, \lambda_{j_2})$ to represent some constants that are not depending on $h, T, B$.

The expectation $\mathcal{I}_1\left(M, h, T, \lambda_{l_1}, \lambda_{j_1}, \lambda_{l_2}, \lambda_{j_2}, \alpha\right)$ is computed exactly the same way as in the proof of Theorem 1 by using the expectation results Equation (31) and Equation (32). Notice that the expectation result Equation (31) (when $s = t$) has the same order in $t$ as the expectation Equation (15). Moreover, the two expectations Equation (32) and Equation (17) have the same orders in $s$ and $t$. Thus, $\mathcal{I}_1\left(M, h, T, \lambda_{l_1}, \lambda_{j_1}, \lambda_{l_2}, \lambda_{j_2}, \alpha\right)$ has the same orders in $h, T, B$ as the scalar MSE, i.e.

$$\mathcal{I}_1\left(M, h, T, \lambda_{l_1}, \lambda_{j_1}, \lambda_{l_2}, \lambda_{j_2}, \alpha\right) = \left(\bar{C}_1 + C_1\left(\lambda_{l_1}, \lambda_{j_1}, \lambda_{l_2}, \lambda_{j_2}\right)\mathcal{O}(T)\right)T^2 h^2 + \mathcal{O}(h^3)$$

$$+ \left(\bar{C}_2 + C_2\left(\lambda_{l_1}, \lambda_{j_1}, \lambda_{l_2}, \lambda_{j_2}\right)\mathcal{O}(T)\right)\frac{T^5}{hB} + \mathcal{O}\left(\frac{1}{B}\right)$$

The expectation $\mathcal{I}_2\left(M, h, T, \lambda_{l_1}, \lambda_{j_1}, \lambda_{l_2}, \lambda_{j_2}, \alpha, \beta\right)$ can be computed directly and has the result:

$$\mathcal{I}_2\left(M, h, T, \lambda_{l_1}, \lambda_{j_1}, \lambda_{l_2}, \lambda_{j_2}, \alpha, \beta\right) = \frac{\left(e^{(\lambda_{l_1}+\lambda_{j_1})T} - 1\right)\left(e^{(\lambda_{l_2}+\lambda_{j_2})T} - 1\right)h^2}{4\left(\lambda_{l_1} + \lambda_{j_1}\right)\left(\lambda_{l_2} + \lambda_{j_2}\right)} + \mathcal{O}(h^3)$$

$$= \left(\frac{1}{4}T^2 + C_3(\lambda_{l_1}, \lambda_{j_1}, \lambda_{l_2}, \lambda_{j_2})\mathcal{O}(T^3))\right)h^2 + \mathcal{O}(h^3).$$

The expectation $\mathcal{I}_3\left(M, h, T, \lambda_{l_1}, \lambda_{j_1}, \lambda_{l_2}, \lambda_{j_2}, \alpha, \beta\right)$ can be computed as follows:

$$
\mathcal{I}_3\left(M, h, T, \lambda_{l_1}, \lambda_{j_1}, \lambda_{l_2}, \lambda_{j_2}, \alpha, \beta\right)
$$

$$
= \frac{h^2}{M} \sum_{k=0}^{n} \frac{\left(e^{\left(\lambda_{l_1}+\lambda_{l_2}\right)kh} - 1\right)\left(e^{\left(\lambda_{j_1}+\lambda_{j_2}\right)kh} - 1\right)h^2}{\left(\lambda_{l_1}+\lambda_{l_2}\right)\left(\lambda_{j_1}+\lambda_{j_2}\right)} +
$$

$$
\frac{h^2}{M} \sum_{k<q} \frac{e^{\lambda_{l_1}kh + \lambda_{l_2}qh + \lambda_{j_1}kh + \lambda_{j_2}qh}}{\left(\lambda_{l_1}+\lambda_{l_2}\right)\left(\lambda_{j_1}+\lambda_{j_2}\right)} \left(1 - e^{-\left(\lambda_{l_1}+\lambda_{l_2}\right)kh}\right)\left(1 - e^{-\left(\lambda_{j_1}+\lambda_{j_2}\right)kh}\right)
$$

$$
\frac{h^2}{M} \sum_{k<q} \frac{e^{\lambda_{l_1}qh + \lambda_{l_2}kh + \lambda_{j_1}qh + \lambda_{j_2}kh}}{\left(\lambda_{l_1}+\lambda_{l_2}\right)\left(\lambda_{j_1}+\lambda_{j_2}\right)} \left(1 - e^{-\left(\lambda_{l_1}+\lambda_{l_2}\right)kh}\right)\left(1 - e^{-\left(\lambda_{j_1}+\lambda_{j_2}\right)kh}\right)
$$

$$
= \left(\bar{C}_4 + C_4\left(\lambda_{l_1}, \lambda_{j_1}, \lambda_{l_2}, \lambda_{j_2}\right)\mathcal{O}(T)\right)\frac{T^5}{hB} + \mathcal{O}\left(\frac{1}{B}\right).
$$

Thus, the final result is obtained by the expression of MSE in Equation (30), Equation (34) and the above computations. Again, we rely on symbolic computation to obtain the expression and corresponding Taylor approximations and include the notebooks of all derivations in the supplementary material. □

The extension from Theorem 3.4 to the discounted finite-horizon results can be done in the same way as in the above proof (add the discount factor $\gamma$ in $\hat{V}_M$) by using the expectation cost for any $\lambda_1$ and $\lambda_2$:

$$
\int_0^T \gamma^t \mathbb{E}\left[\int_0^t e^{\lambda_1(t-u)}\, \mathrm{d}w(u) \int_0^s e^{\lambda_2(-u)}\, \mathrm{d}w(u)\right]\, \mathrm{d}t
$$

$$
= \frac{1}{\left(\lambda_1+\lambda_2\right)}\left(\frac{\gamma^T e^{\left(\lambda_1+\lambda_2\right)T} - 1}{\log\left(\gamma\right) + \left(\lambda_1+\lambda_2\right)} - \frac{\gamma^T - 1}{\log\left(\gamma\right)}\right).
$$

## C.2   Proof of Corollary 3.7

*Proof.* We shall follow the similar proof as in the proof of Theorem 3.4 and the proof of Theorem 3.6.

Continuing from Equation (34), in infinite-horizon discounted setting, we have

$$
\mathcal{I}_1\left(M, h, T, \lambda_{l_1}, \lambda_{j_1}, \lambda_{l_2}, \lambda_{j_2}, \gamma, \alpha\right)
$$

$$
= \mathbb{E}\Bigg\{\left[\frac{h}{M}\sum_{i=1}^{M}\sum_{k=0}^{N-1}\gamma^{kh}\left(\int_0^{kh} e^{\lambda_{l_1}(kh-s)}\, \mathrm{d}w_\alpha^{(i)}(s)\right)\left(\int_0^{kh} e^{\lambda_{j_1}(kh-s)}\, \mathrm{d}w_\alpha^{(i)}(s)\right)\right.
$$

$$
\left. - \int_0^\infty \gamma^t \mathbb{E}\left[\left(\int_0^t e^{\lambda_{l_1}(t-s)}\, \mathrm{d}w_\alpha(s)\right)\left(\int_0^t e^{\lambda_{j_1}(t-s)}\, \mathrm{d}w_\alpha(s)\right)\right]\, \mathrm{d}t\right] \times
$$

$$
\left[\frac{h}{M}\sum_{i=1}^{M}\sum_{k=0}^{N-1}\gamma^{kh}\left(\int_0^{kh} e^{\lambda_{l_2}(kh-s)}\, \mathrm{d}w_\alpha^{(i)}(s)\right)\left(\int_0^{kh} e^{\lambda_{j_2}(kh-s)}\, \mathrm{d}w_\alpha^{(i)}(s)\right)\right.
$$

$$
\left.- \int_0^\infty \gamma^t \mathbb{E}\left[\left(\int_0^t e^{\lambda_{l_2}(t-s)}\, \mathrm{d}w_\alpha(s)\right)\left(\int_0^t e^{\lambda_{j_2}(t-s)}\, \mathrm{d}w_\alpha(s)\right)\right]\, \mathrm{d}t\right]\Bigg\},
$$

and

$$
\mathcal{I}_2\left(M, h, T, \lambda_{l_1}, \lambda_{j_1}, \lambda_{l_2}, \lambda_{j_2}, \gamma, \alpha, \beta\right)
$$

$$
= \left[\frac{h}{\left(\lambda_{l_1}+\lambda_{j_1}\right)}\left(\frac{1 - \gamma^T e^{\left(\lambda_{l_1}+\lambda_{j_1}\right)T}}{1 - \gamma^h e^{\left(\lambda_{l_1}+\lambda_{j_1}\right)h}} - \frac{1 - \gamma^T}{1 - \gamma^h}\right) - \frac{1}{\left(\lambda_{l_1}+\lambda_{j_1}\right)}\left(\frac{1}{\log\left(\gamma\right)} - \frac{1}{\log\left(\gamma\right) + \lambda_{l_1} + \lambda_{j_1}}\right)\right]
$$

$$
\times \left[\frac{h}{\left(\lambda_{l_2}+\lambda_{j_2}\right)}\left(\frac{1 - \gamma^T e^{\left(\lambda_{l_2}+\lambda_{j_2}\right)T}}{1 - \gamma^h e^{\left(\lambda_{l_2}+\lambda_{j_2}\right)h}} - \frac{1 - \gamma^T}{1 - \gamma^h}\right) - \frac{1}{\left(\lambda_{l_2}+\lambda_{j_2}\right)}\left(\frac{1}{\log\left(\gamma\right)} - \frac{1}{\log\left(\gamma\right) + \lambda_{l_2} + \lambda_{j_2}}\right)\right],
$$

and

$$\mathcal{I}_3\left(M, h, T, \lambda_{l_1}, \lambda_{j_1}, \lambda_{l_2}, \lambda_{j_2}, r, \alpha, \beta\right)$$
$$= \mathbb{E}\left\{\left[\frac{h}{M}\sum_{i=1}^{M}\sum_{k=0}^{N-1}\gamma^{kh}\left(\int_0^{kh}e^{\lambda_{l_1}(kh-s)}\,\mathrm{d}w_\alpha^{(i)}(s)\right)\left(\int_0^{kh}e^{\lambda_{j_1}(kh-s)}\,\mathrm{d}w_\beta^{(i)}(s)\right)\right]\times\right.$$
$$\left.\left[\frac{h}{M}\sum_{i=1}^{M}\sum_{k=0}^{N-1}\gamma^{kh}\left(\int_0^{kh}e^{\lambda_{l_2}(kh-s)}\,\mathrm{d}w_\alpha^{(i)}(s)\right)\left(\int_0^{kh}e^{\lambda_{j_2}(kh-s)}\,\mathrm{d}w_\beta^{(i)}(s)\right)\right]\right\}.$$

Similar arguments as in proof of Theorem 3.4, we can conclude $\mathcal{I}_1\left(M, h, T, \lambda_{l_1}, \lambda_{j_1}, \lambda_{l_2}, \lambda_{j_2}, \alpha\right)$ has the same orders in $h, B, T$ as the MSE result in Theorem 3.6.

Moreover, let $C_i(\lambda_{l_1}, \lambda_{j_1}, \lambda_{l_2}, \lambda_{j_2}, \gamma, T)$'s are some constants that depend on $\lambda_{l_1}, \lambda_{j_1}, \lambda_{l_2}, \lambda_{j_2}, \gamma, T$, then

$$\mathcal{I}_2\left(M, h, T, \lambda_{l_1}, \lambda_{j_1}, \lambda_{l_2}, \lambda_{j_2}, \gamma, \alpha, \beta\right)$$
$$= \sigma^4\gamma^{2T}\left(C_1(\lambda_{l_1}, \lambda_{j_1}, \lambda_{l_2}, \lambda_{j_2}, \gamma, T) + C_2(\lambda_{l_1}, \lambda_{j_1}, \lambda_{l_2}, \lambda_{j_2}, \gamma, T)h\right)$$
$$+ \sigma^4\gamma^{T}\left(C_3(\lambda_{l_1}, \lambda_{j_1}, \lambda_{l_2}, \lambda_{j_2}, \gamma, T)h^2 + C_4(\lambda_{l_1}, \lambda_{j_1}, \lambda_{l_2}, \lambda_{j_2}, \gamma, T)h^3\right)$$
$$+ \sigma^4\left(\frac{1}{144} + \gamma^{T}C_5(\lambda_{l_1}, \lambda_{j_1}, \lambda_{l_2}, \lambda_{j_2}, \gamma, T)\right)h^4 + \mathcal{O}(h^5),$$

and

$$\mathcal{I}_3\left(M, h, T, \lambda_{l_1}, \lambda_{j_1}, \lambda_{l_2}, \lambda_{j_2}, \gamma, \alpha, \beta\right)$$
$$= \frac{h^2}{M}\sum_{k=0}^{N-1}\frac{\left(e^{(\lambda_{l_1}+\lambda_{l_2})kh}-1\right)\left(e^{(\lambda_{j_1}+\lambda_{j_2})kh}-1\right)h^2\gamma^{2kh}}{(\lambda_{l_1}+\lambda_{l_2})(\lambda_{j_1}+\lambda_{j_2})}+$$
$$\frac{h^2}{M}\sum_{k<q}\frac{e^{\lambda_{l_1}kh+\lambda_{l_2}qh+\lambda_{j_1}kh+\lambda_{j_2}qh}}{(\lambda_{l_1}+\lambda_{l_2})(\lambda_{j_1}+\lambda_{j_2})}\left(1-e^{-(\lambda_{l_1}+\lambda_{l_2})kh}\right)\left(1-e^{-(\lambda_{j_1}+\lambda_{j_2})kh}\right)\gamma^{(k+q)h}$$
$$\frac{h^2}{M}\sum_{k<q}\frac{e^{\lambda_{l_1}qh+\lambda_{l_2}kh+\lambda_{j_1}qh+\lambda_{j_2}kh}}{(\lambda_{l_1}+\lambda_{l_2})(\lambda_{j_1}+\lambda_{j_2})}\left(1-e^{-(\lambda_{l_1}+\lambda_{l_2})kh}\right)\left(1-e^{-(\lambda_{j_1}+\lambda_{j_2})kh}\right)\gamma^{(k+q)h}$$
$$= C_6\left(\lambda_{l_1}, \lambda_{j_1}, \lambda_{l_2}, \lambda_{j_2}, \gamma, T\right)\frac{T^5}{hB} + \mathcal{O}\left(\frac{1}{B}\right).$$

The result in this Corollary is obtained by combining the above results. And we include the notebooks of all derivations in the supplementary material. □

## C.3 The case when $A$ is a general stable matrix

**Lemma C.1** (MSE when $A$ is a general stable matrix ). *Let $A$ be a stable $n \times n$ matrix with distinct eigenvalues $\lambda_1, \cdots, \lambda_m$ and corresponding multiplicities $q_1, \cdots, q_m$. There exist some constants $\{\bar{C}_i\}_{i=1}^m$, $\bar{C}_0$ and $C_j(\lambda_1, \cdots, \lambda_m, \gamma, T)$'s, such that the mean-squared error of the Monte-Carlo estimator in different setting satisfies*

*(1) Finite-Horizon undiscounted setting:*

$$MSE_T \in \left[\sum_{i=1}^{m}q_i\bar{C}_i MSE_T(h, B, \lambda_i), \quad C_1(\lambda_1, \cdots, \lambda_m, T)\sigma^4 T^2 h^2\right.$$
$$\left. + \frac{\left(\bar{C}_2 + C_3(\lambda_1, \cdots, \lambda_m, T)\mathcal{O}(T)\right)\sigma^4 T^{2n+3}}{Bh} + \mathcal{O}(h^3) + \mathcal{O}(\frac{1}{B})\right], \qquad (35)$$

*where $MSE_T(h, B, \lambda_i)$ is the mean-squared error of the Monte-Carlo estimator in Theorem 3.1 by replacing the drift $a$ by $\lambda_i$.*

*(2) Finite-Horizon discounted setting:*

$$MSE_T \in \Bigg[ \sum_{i=1}^m q_i \bar{C}_i MSE_T(h, B, \gamma, \lambda_i), \quad C_4(\lambda_1, \cdots, \lambda_m, \gamma, T)\sigma^4 \gamma^{2T} T^2 h^2$$

$$+ C_5(\lambda_1, \cdots, \lambda_m, \gamma, T)\sigma^4 \gamma^T h^3 + C_6(\lambda_1, \cdots, \lambda_m, T)\sigma^4 h^4$$

$$+ \frac{(C_7(\lambda_1, \cdots, \lambda_m, \gamma, T))\sigma^4 T^{2n-1}}{Bh} + \mathcal{O}(h^5) + \mathcal{O}(\frac{1}{B}) \Bigg], \tag{36}$$

*where $MSE_T(h, B, \gamma, \lambda_i)$ is the mean-squared error of the Monte-Carlo estimator in Lemma B.1 by replacing the drift $a$ by $\lambda_i$.*

*(3) Infinite-Horizon discounted setting:*

$$MSE_\infty \in \Bigg[ \sum_{i=1}^m q_i \bar{C}_i MSE_\infty(h, B, \gamma, \lambda_i),$$

$$(C_8(\lambda_1, \cdots, \lambda_m, \gamma, T) + C_9(\lambda_1, \cdots, \lambda_m, \gamma, T)h)\sigma^4 \gamma^{2T}$$

$$+ \left( C_{10}(\lambda_1, \cdots, \lambda_m, \gamma, T)h^2 + C_{11}(\lambda_1, \cdots, \lambda_m, \gamma, T)h^3 \right)\sigma^4 \gamma^T$$

$$+ C_{12}(\lambda_1, \cdots, \lambda_m, T)\sigma^4 h^4 + \frac{(C_{13}(\lambda_1, \cdots, \lambda_m, \gamma, T))\sigma^4 T^{2n-1}}{Bh}$$

$$+ \mathcal{O}(h^5) + \mathcal{O}(\frac{1}{B}) \Bigg], \tag{37}$$

*where $MSE_\infty(h, B, \gamma, \lambda_i)$ is the mean-squared error of the Monte-Carlo estimator in Theorem 3.6 by replacing the drift $a$ by $\lambda_i$.*

*Proof.* As we can see the proof of Lemma B.1 is based on the proof of Theorem 3.1 with adding a discount factor $\gamma$, and the proof of Theorem 3.6 is based on the proof of Lemma B.1 with the decomposition Equation (12). By using the same flow direction, it is sufficient to show the result in case (1) and the results in case (2) and (3) follows.

Consider the decomposition of $MSE_T$ in finite-horizon undiscounted setting:

$$MSE_T = \mathbb{E}\left[ (\hat{V}_M - V_T)^2 \right]$$

$$= \mathbb{E}\left[ \left( \hat{V}_M - \mathbb{E}\left[ \hat{V}_M \right] + \mathbb{E}\left[ \hat{V}_M \right] - V_T \right)^2 \right]$$

$$= \underbrace{\mathbb{E}\left[ \hat{V}_M^2 \right] - \mathbb{E}\left[ \hat{V}_M \right]^2}_{\text{Part1}} + \underbrace{\left( \mathbb{E}\left[ \hat{V}_M \right] - V_T \right)^2}_{\text{Part2}}$$

Before the analysis of part 1 and part 2, we will introduce the following mean-squared error notations for the finite-horizon undiscounted scalar case with drift $\lambda_i$:

$$MSE_T(h, B, \lambda_i) = \text{Var}(h, \lambda_i) + \text{Approximation}(h, B, \lambda_i), \tag{38}$$

where $\text{Var}(h, \lambda_i) = \mathbb{E}\left[ \hat{V}_M^2 \right] - \mathbb{E}\left[ \hat{V}_M \right]^2$ and $\text{Approximation}(h, B, \lambda_i) = \left( \mathbb{E}\left[ \hat{V}_M \right] - V_T \right)^2$.

For part 1:

$$\mathbb{E}\left[ \hat{V}_M^2 \right] = \frac{h^2}{M} \sum_{i,j,k,l} \mathbb{E}\left[ X_i(kh)^\top Q X_i(kh) X_j(lh)^\top Q X_j(lh) \right]$$

$$= \frac{h^2}{M^2} \sum_{i,j,k,l} \left[ \mathbb{E}\left[ X_i(kh)^\top Q X_i(kh) \right] \mathbb{E}\left[ X_j(lh)^\top Q X_j(lh) \right] + 2\text{tr}\left\{ Q \mathbb{E}\left[ X_i(kh) X_j(lh)^\top \right] \right\}^2 \right]$$

$$= h^2 \sum_{k,l} \mathbb{E}\left[ X(kh)^\top Q X(kh) \right] \mathbb{E}\left[ X_j(lh)^\top Q X(lh) \right] +$$

$$\frac{2h^2}{M} \sum_k \text{tr}\left\{ Q \mathbb{E}\left[ X(kh) X(kh)^\top \right] \right\}^2 + \frac{4h^2}{M} \sum_{k<l} \text{tr}\left\{ Q \mathbb{E}\left[ X(kh) X(lh)^\top \right] \right\}^2, \tag{39}$$

where the second equality is based on Isserlis' theorem and the trace operation.

Notice that $\mathbb{E}\left[\hat{V}_M\right]^2 = h^2 \sum_{k,l} \mathbb{E}\left[X(kh)^\top Q X(kh)\right]\mathbb{E}\left[X(lh)^\top Q X(lh)\right]$, thus

$$\mathbb{E}\left[\hat{V}_M^2\right] - \mathbb{E}\left[\hat{V}_M\right]^2$$
$$= \frac{2h^2}{M}\sum_k \operatorname{tr}\left\{Q\mathbb{E}\left[X(kh)X(kh)^\top\right]\right\}^2 + \frac{4h^2}{M}\sum_{k<l}\operatorname{tr}\left\{Q\mathbb{E}\left[X(kh)X(lh)^\top\right]\right\}^2$$

To analyze the above form, we decompose the matrix $A$ by it Jordan form, i.e. $A = P^{-1}JP$ for some inevitable matrix $P$ and $J = \operatorname{diag}(J_i, \cdots, J_m)$, where $J_i$ is the Jordan block corresponding to the eigenvalue $\lambda_i$.

Notice that $e^{J(kh-s)} = \operatorname{diag}(e^{J_1(kh-s)}, \cdots, e^{J_m(kh-s)})$, where

$$e^{J_i(kh-s)} = e^{\lambda_i(kh-s)}\begin{pmatrix} 1 & kh-s & \frac{(kh-s)^2}{2!} & \cdots & \frac{(kh-s)^{q_i-1}}{(q_i-1)!} \\ & 1 & kh-s & \cdots & \frac{(kh-s)^{q_i-2}}{(q_i-2)!} \\ & & \ddots & & \vdots \\ & & & & 1 \end{pmatrix}.$$

Combining with the fact that for any $k, l$,

$$\mathbb{E}\left[X(kh)X(lh)^\top\right] = \int_0^{kh\wedge lh} e^{A(kh-s)}e^{A^\top(lh-s)}\,\mathrm{d}s$$
$$= \int_0^{kh\wedge lh} P^{-1}e^{J(kh-s)}PP^\top e^{J^\top(lh-s)}P^{-\top}\,\mathrm{d}s\,,$$

we can conclude that for any $k \leq l$, $\operatorname{tr}\left\{Q\mathbb{E}\left[X(kh)X(lh)^\top\right]\right\}$ is a linear combination of $\mathcal{L}_{1,i,j}$ and $\mathcal{L}_{2,i,j}$ for all $i, j$, where

$$\mathcal{L}_{1,i,j} := C_{1,i,j}\int_0^{kh} e^{(\lambda_i(kh-s)+\lambda_j(lh-s))}\,\mathrm{d}s$$
$$= C_{1,i,j}\frac{e^{\lambda_i kh}+e^{\lambda_j}}{\lambda_i+\lambda_j}\left(1 - e^{-(\lambda_i+\lambda_j)kh}\right)$$
$$\mathcal{L}_{2,i,j} := C_{2,i,j}\int_0^{kh} e^{(\lambda_i(kh-s)+\lambda_j(lh-s))}(kh-s)^{\tilde{q}_i}(lh-s)^{\tilde{q}_j}\,\mathrm{d}s\,,$$

where $C_{1,i,j}$, $C_{i,j}$ are some constants and $\tilde{q}_i \in \{0, \cdots, q_i-1\}$, $\tilde{q}_j \in \{0, \cdots, q_j-1\}$.

For the integral in $\mathcal{L}_{2,i,j}$, as $\tilde{q}_i + \tilde{q}_j \leq n-1$, we can have the inequality:

$$\int_0^{kh} e^{(\lambda_i(kh-s)+\lambda_j(lh-s))}(kh-s)^{\tilde{q}_i}(lh-s)^{\tilde{q}_j}\,\mathrm{d}s$$
$$\leq T^{n-1}\int_0^{kh} e^{(\lambda_i(kh-s)+\lambda_j(lh-s))}\,\mathrm{d}s\,. \tag{40}$$

Since

$$\operatorname{tr}\left\{Q\mathbb{E}\left[X(kh)X(lh)^\top\right]\right\}^2 = \sum_{i_1,j_1,i_2,j_2}\sum_{k,l}\prod_{l_1,l_2\in\{1,2\}}\mathcal{L}_{l_1,i_1,j_1}\mathcal{L}_{l_2,i_2,j_2}\,,$$

and all the terms are nonnegative. We drop all terms that include $\mathcal{L}_{2,i,j}$ factor and only include the $\mathcal{L}_{1,i,i}^2$ with $k = l$ terms in the lower bound of part 1. That is to say, the lower bound of part 1 is $\sum_{i=1}^m q_i \bar{C}_i \operatorname{Var}(h, \lambda_i)$.

The upper bound of part 1 can be obtained by replacing all $\mathcal{L}_{1,i,j}$ factors by $\mathcal{L}_{2,i,j}$ and use the bound given in Equation (40). This leads to the upper bound for part 1 is $\frac{(\bar{C}_2+C_3(\lambda_1,\cdots,\lambda_m,T)\mathcal{O}(T))\sigma^4 T^{2n+5}}{Bh} + \mathcal{O}(\frac{1}{B})$.

For Part 2, let $g(t) = \mathbb{E}\left[X(t)^\top Q X(t)\right]$ on $[0, T]$. Then $\mathbb{E}\left[\hat{V}_M\right]$ is the left Riemann sum approximation of $g(t)$, by the property of Riemann approximation,

$$\left|\mathbb{E}\left[\hat{V}_M\right] - V_T\right| \approx 2hTg(T) + \mathcal{O}(h^2),$$

where

$$g(T) = \mathrm{tr}\left\{Q\mathbb{E}\left[X(T)X(T)^\top\right]\right\} = \sigma^2 \mathrm{tr}\left\{Q\int_0^T e^{A(t-s)}e^{A^\top(t-s)}\,\mathrm{d}s\right\},$$

which is a constant depends on $\lambda_1, \cdots, \lambda_m, T$. Thus

$$\left(\mathbb{E}\left[\hat{V}_M\right] - V_T\right)^2 \approx C_1(\lambda_1, \cdots, \lambda_m, T)\sigma^4 T^2 h^2 + \mathcal{O}(h^3),$$

which has the same order in $h$ as the scalar case in finite-horizon undiscounted setting. Thus the result in Equation (35) is obtained by combining part 1 bounds and part 2 approximation.

As we explained in the beginning of this proof, in the finite-horizon discounted setting, we will follow the similar arguments as in the proof of Equation (35) to obtain result Equation (36).

For the infinite-horizon discounted setting, the corresponding part 1 in the $\mathrm{MSE}_\infty$ is the same as the part 1 in $\mathrm{MSE}_T$ of Equation (36). The part 2 is approximated by using the decomposition Equation (12) and the fact that

$$V_{t,\infty} = \int_T^\infty \gamma^t \mathbb{E}\left[X(t)^\top Q X(t)\right]\,dt = \gamma^T C(r, T, \lambda_1, \cdots, \lambda_m).$$

To verify $V_{T,\infty}$ is $\mathcal{O}(\gamma^T)$, one can find the bounds of $V_{T,\infty}$ by using the similar arguments in the above proof of Equation (35) and the following inequality:

$$\int_0^t e^{(\lambda_i+\lambda_j)(t-s)}(t-s)^{\tilde{q}_i+\tilde{q}_j}\,\mathrm{d}s$$

$$\leq t^{n-1}\int_0^t e^{(\lambda_i+\lambda_j)(t-s)}\,\mathrm{d}s = \frac{t^{n-1}}{(\lambda_i+\lambda_j)}\left(e^{(\lambda_i+\lambda_j)t} - 1\right).$$

Then the components in $\int_T^\infty \gamma^t \mathbb{E}\left[X(t)X(t)^\top\right]\,dt$ is lower bounded by $\int_T^\infty \frac{r^t}{(\lambda_i+\lambda_j)}\left(e^{(\lambda_i+\lambda_j)t} - 1\right)\,\mathrm{d}t$ and upper bounded by $\int_T^\infty \frac{r^t t^{n-1}}{(\lambda_i+\lambda_j)}\left(e^{(\lambda_i+\lambda_j)t} - 1\right)\,\mathrm{d}t$. By the celebrated approximation of incomplete gamma function when $T$ is large, we have

$$\int_T^\infty \frac{r^t t^{n-1}}{(\lambda_i+\lambda_j)}\left(e^{(\lambda_i+\lambda_j)t} - 1\right)\,\mathrm{d}t \approx \frac{r^T T^{n-1}}{(\lambda_i+\lambda_j)}\left(e^{(\lambda_i+\lambda_j)T} - 1\right).$$

Followed by

$$V_{T,\infty} = \mathrm{tr}\left\{Q\int_T^\infty \gamma^t \mathbb{E}\left[X(t)X(t)^\top\right]\,\mathrm{d}t\right\},$$

we can obtain that $V_{T,\infty} = \gamma^T C(r, T, \lambda_1, \cdots, \lambda_m)$.

This result leads to the fact that part 2 is

$$\left(C_8(\lambda_1, \cdots, \lambda_m, \gamma, T) + C_9(\lambda_1, \cdots, \lambda_m, \gamma, T)h\right)\sigma^4\gamma^{2T} + \left(C_{10}(\lambda_1, \cdots, \lambda_m, \gamma, T)h^2\right.$$
$$\left. + C_{11}(\lambda_1, \cdots, \lambda_m, \gamma, T)h^3\right)\sigma^4\gamma^T + C_{12}(\lambda_1, \cdots, \lambda_m, T)\sigma^4 h^4 + \mathcal{O}(h^5),$$

which coincides with the $\mathrm{Var}(h\lambda_i)$ in the infinite-horizon discounted scalar case. The results in (3) then follows. $\qquad\square$

# D Complements to Numerical Simulations

The LQR experiments were run on a MacBook pro with an i9 CPU and 16GB of RAM.

### D.1 Randomly-sampled Matrices

The matrices $A$ in Figure 1(c) and Figure 1(d), corresponding to controlled multi-dimensional linear systems, are generated according to the procedure described below. It ensures the stability of the resulting system, i.e., that eigenvalues of the sampled matrix are negative.

The procedure works by random sampling an eigendecomposition. It starts with uniformly sampling two eigenvalues from disjoint, bounded intervals, $\lambda_1 \in [-1.5, -1.0)$ and $\lambda_3 \in (-1.0, -0.75]$. The final eigenvalue is set to be $\lambda_2 = -1.0$. Note that since all the eigenvalues sampled are negative, any matrix whose eigenvalues are $\lambda_1, \lambda_2, \lambda_3$ is said to be stable. Next, the eigenvectors are sampled randomly from the classical compact groups detailed in [Mezzadri, 2006]. For that, we randomly sample an orthogonal matrix $L$ using a built-in SCIPY [Virtanen et al., 2020] routine, ORTHO_GROUP.RVS. Now let $\Lambda = \text{diag}(\lambda_1, \lambda_2, \lambda_3)$, the random, dense, stable matrix $A$ is obtained by computing the product $A = L^\top \Lambda L$.

### D.2 Trade-off in LQR with Scaled Identity Matrices

In order to better understand the transition from scalar to vector case in the trade-off of the step-size due to the MSE in Section 3, numerical experiments for the case of identity matrices scaled by a constant are provided in this section. This allows to characterise the role played by the eigenvalues, with respect to the parameter $a$ in the scalar case.

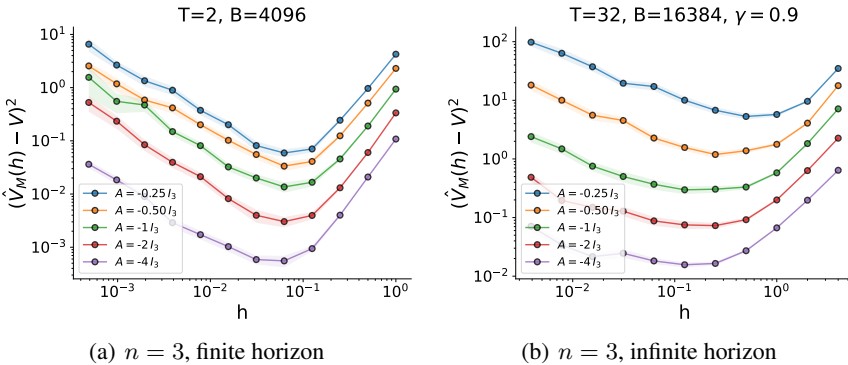

(a) $n = 3$, finite horizon          (b) $n = 3$, infinite horizon

Figure 4: Mean-squared error trade-off in LQR with scaled identity matrices A. The plots show the dependence of the optimal step-size on the eigenvalues of the linear systems in both finite and infinite horizon settings. The same trend of the scalar case w.r.t. the parameter $a$ can be observed here.

As expected, results in Figure 4 suggest that the trade-off for scaled identity matrices is very similar to the one in the scalar case. In this simple case the eigenvalues that are identical on all dimensions play the same role as the parameter $a$, i.e., by decreasing them, the trade-off shifts towards a smaller value for the optimal step-size, as suggested also by Figure 1(b).

### D.3 Comparison of Empirical and Analytical MSEs in One-dimensional Langevin Systems

Figure 5 compares the empirical MSE in Figures 1(a) and 1(b) with the analytical MSEs, including both the exact and approximate versions. We can observe that the empirical results are consistent with the analysis. The error in approximate MSE becomes noticeable for large $h$ especially when $h > 1$, as expected. Nevertheless, the optimal $h^*$ remains consistent.

### D.4 Implementation Details for Nonlinear Systems Experiments

We summarize the environment-specific parameters in Table 1 for the nonlinear-system experiments.

**Compute Resources**    For training the stable policy for non-mujoco environments, we used a server with one GTX 1080. The training for MuJoCo environments was conducted on a cluster, using a single V100 Volta for each environment. For inference, since we need to run many episodes, we scaled things up on a cluster of CPUs.

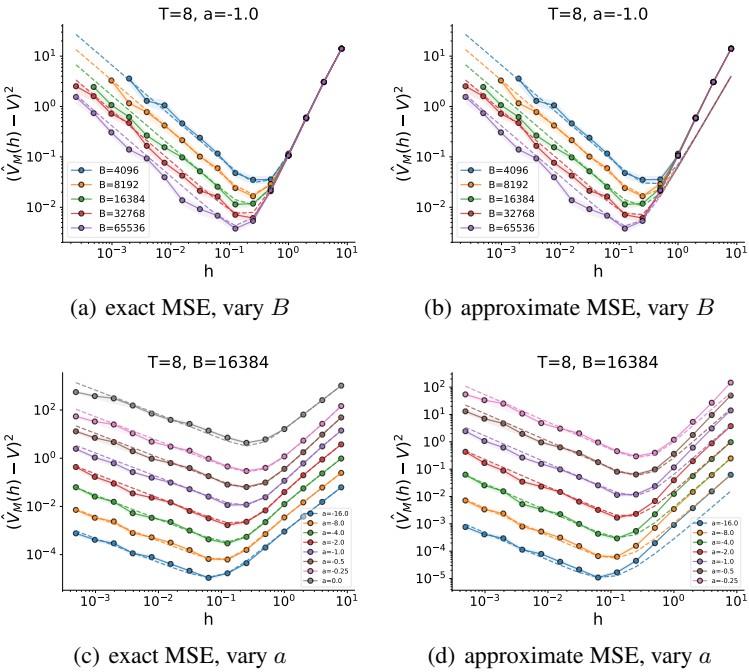

Figure 5: Comparison between the empirical (solid) and analytical MSEs (dashed) in one-dimensional Langevin systems.

**Training Details**   We used the CDAU algorithm described in [Tallec et al., 2019] to train the policies since we find this algorithm to be robust to time discretization, especially when the environment runs at $\delta t = 0.001$. We closely followed the hyper-parameters setup described in Section 2 in the Appendix of [Tallec et al., 2019]. For more details, please refer to their paper, and the code in the supplementary materials.

**Inference Details**   We run the policy for $300k$ episodes at the finest time discretization $\delta t = 0.001$ ($600k$ in InvertedDoublePendulum and Pusher) and store the reward sequences. The number of episodes is chosen to be sufficient for 30 runs and the largest possible number of trajectories (depending on $h$ and the data budget $B$). These data get down-sampled offline for different choices of $h$. The episodes are randomly shuffled when we vary $B$.

Table 1: The setup of the environments. *: In MuJoCo environments in OpenAI Gym, $\delta t =$ timestep $*$ frame_skip, where 'timestep' (the step size of the MuJoCo dynamics simulation) and 'frame_skip' (the algorithmic step size) are two pre-set quantities in their implementation. In our setup, $\delta t = 0.001$ seconds for the proxies to the continuous-time environments.

| Environment | Episode Length (steps) | Original* $\delta t$ | Horizon $T$ (seconds) |
|---|---|---|---|
| Pendulum | 200 | 0.05 | 10 |
| BipedalWalker | 500 | 0.02 | 10 |
| InvertedDoublePendulum | 1000 | 0.05 | 50 |
| Pusher | 1000 | 0.05 | 50 |
| Swimmer | 1000 | 0.04 | 40 |
| Hopper | 1000 | 0.008 | 8 |
| HalfCheetah | 1000 | 0.05 | 50 |
| Ant | 1000 | 0.05 | 50 |

| Environment | $B_0$ | $h$ |
|---|---|---|
| Pendulum | $10k$ | [0.001, 0.002, 0.004, 0.01, 0.02, 0.04, 0.1] |
| BipedalWalker | $10k$ | [0.001, 0.002, 0.004, 0.01, 0.02, 0.04, 0.1] |
| InvertedDoublePendulum | $25k$ | [0.002, 0.004, 0.01, 0.02, 0.04, 0.1, 0.2, 0.4, 1] |
| Pusher | $25k$ | [0.002, 0.004, 0.01, 0.02, 0.04, 0.1, 0.2, 0.4, 1] |
| Swimmer | $20k$ | [0.002, 0.004, 0.01, 0.02, 0.04, 0.1] |
| Hopper | $8k$ | [0.001, 0.002, 0.004, 0.01, 0.02, 0.04, 0.1] |
| HalfCheetah | $25k$ | [0.002, 0.004, 0.01, 0.02, 0.04, 0.1, 0.2, 0.4] |
| Ant | $25k$ | [0.002, 0.004, 0.01, 0.02, 0.04, 0.1, 0.2, 0.4] |

