# OpenReview forum: "Managing Temporal Resolution in Continuous Value Estimation: A Fundamental Trade-off"
_NeurIPS.cc/2023/Conference — NeurIPS 2023 poster_

### Official Review · Reviewer_sR6e · 2023-07-05

**Soundness:** 4 excellent
**Presentation:** 3 good
**Contribution:** 2 fair
**Rating:** 6
**Confidence:** 3

**Summary:**

The paper discusses the impact of time discretization on estimating cost functions for continuous-time stochastic optimal control problems. The authors consider a discrete-time Monte Carlo estimator based on the left-rule formula for numerical integration. Under the LQG assumption, the authors derive closed-form results for the mean square error (MSE), which depend on the sample size and the integration step size. The authors provide a numerical evaluation based on their modeling assumptions and propose hypotheses on how to extend their findings to general nonlinear systems.

**Strengths:**

A large class of problems is naturally described as continuous-time systems, making investigations on this topic important. The paper effectively demonstrates how time discretization impacts the estimation of continuous-time quantities, such as the cost function. The work is well written and easy to follow. The results derived in this work are well presented and properly justified.

**Weaknesses:**

I believe the main weakness lies in the significance of the contribution itself. It is to me not particularly surprising that closed-form results for MSE under the LQG assumption can be found. Additionally, it is unclear to me when this method should be used. For instance, if one knows they have an LQG problem, they already know that the value function is a quadratic function and its coefficient can be computed by estimating the dynamics parameter.

Furthermore, the numerical evaluation for the nonlinear systems appears somewhat weak, and I am not sure whether the results can be easily transferred from the linear to the nonlinear regime.


**Questions:**

-

**Limitations:**

-

---

> ### Author Rebuttal · Authors · 2023-08-09
>
> We would like to thank the reviewer for the time reviewing our work and for providing valuable feedback.
>
> We respectfully note that the formulation of MSE under a fixed data budget has not been explored before. While it may be reasonable to anticipate a closed-form MSE under this formulation, we believe that the process of devising this formulation and developing an exact characterization of the trade-off w.r.t. the step-size constitutes a significant contribution, even if it appears intuitive in retrospect.\
> Note also that the aim of the paper is not to provide a method for estimating the value-function in the LQG setting, but rather to characterize an overlooked phenomenon affecting any continuous stochastic system.
>
> Our experiments on nonlinear systems in standard benchmark extend beyond the typical scope of experiments in LQG literature. These were designed to assess how well the theoretical findings could still hold when certain assumptions were not met. Interestingly enough, the results seem to extend quite remarkably to this more challenging scenario. This suggests that our findings may have broader applicability than the specific conditions under which our theoretical analysis was established.\
> We hope that this clarifies the purpose and the scope of the evaluation for nonlinear systems, in response to the comment that it “appears somewhat weak”. If it does not address your concern, we kindly invite you to provide further clarification.

---

> > ### Comment · Reviewer_sR6e · 2023-08-21
> >
> > I want to thank the authors for their clarifications. I still believe this is a solid piece of work and a "moderate-to-high impact paper", therefore, I will keep my score.

---

### Official Review · Reviewer_uBm7 · 2023-07-06

**Soundness:** 3 good
**Presentation:** 3 good
**Contribution:** 3 good
**Rating:** 6
**Confidence:** 3

**Summary:**

For the reinforcement learning (RL) setting, many problems are cast into a fixed discrete time sampling of the true underlying system. This paper investigates given some fixed data budget available, what is the optimal sampling rate in terms of temporal resolution to balance the trade-off between approximation and statistical estimation error. Theoretical results are obtained for the exact form of these errors is derived for the MSE of a Monte Carlo in a Langevin process for both finite and infinite horizon setting. This shows that there is an optimal choice in temporal resolution to balance these errors, and this is verified by simulated linear quadratic systems. Similar kind of trade-offs can be found in empirical (nonlinear) environments from gym and MuJoCo.

**Strengths:**

This is a well-written paper which looks at a problem in value estimation that does not appear to have been explicitly studied on this level before. There is detailed theoretical analysis for the case of a Langevin process, and the results are interpreted in a practical way. The demonstration of these theoretical results in numerical simulations in Section 4.1 is very clear. Verification of similar effects on empirical environments is useful to see.

Overall, this paper contributes to the literature on additional considerations that should be made in continuous time RL setting, and it would be useful for future developments in this area.

**Weaknesses:**

Practically the policy is varying at the same time, and perhaps the same step size should not be used over the course of training. Therefore it may be that the practical impact on implementation of this paper may be limited.

For Figure 3, it seems to be misleading to fit a line matching $h = \mathcal{O}(B^{-1/3})$ for the empirical experiments. Particularly given lines like for that of the InvertedDoublePendulum, clearly a different gradient should be considered. The observation on these empirical experiments are interesting, but there is no need to force the results on these nonlinear systems to match the analysis in the previous section.

In setting the step-size $h$, this relies on the ability to perform experiments on a smaller data budget first, which may not always be possible in practice.

**Questions:**

How might an adaptive sampling procedure affect the results?

It is assumed that the estimation horizon is the same length as the episode, this seems like a strong assumption which does not hold in a lot of applications (i.e. you do not know how long the episode is a priori, nor are the episodes the same lengths). How would the results in finite horizon change in this case?

Should the $h$ on line 165 be $h^*$?

Can you clarify where the MSE_T between line 168 and 169 come from?

The second term on the right hand side for the MSE_T has denominator $hB$, will this assume $h = \mathcal{O}(B^{-1/3})$ therefore the second term is $\mathcal{O}(B^{-2/3})$ hence till tends to 0?

**Limitations:**

The theoretical results obtained seems to assume a fixed observation/estimation horizon T, that is pre-set, but since episode are not of fixed lengths in RL settings, and indeed may increase as the policy improves, this may affect the validity of the theoretical results.

---

> ### Author Rebuttal · Authors · 2023-08-10
>
> We would like to thank the reviewer for the time reviewing our work and for providing valuable feedback. We will correct the typos pointed out by the reviewer. In the following, we address the main concerns.
>
> > Can you clarify where the $\text{MSE}_T$ between line $168$ and $169$ come from?
>
> `Re:` It was obtained by plugging $h^*$ from Eq. ($9$) into the $\text{MSE}_T$ in corollary $3.2$.
>
> > The second term on the right hand side for the $\text{MSE}_T$ has denominator $hb$, will this assume $h = \mathcal{O}(B^{-1/3})$ therefore the second term is $\mathcal{O}(B^{-2/3})$ hence till tends to $0$?
>
> `Re:` Note that the expression for the optimal $h$ is indeed $\mathcal{O}(B^{-1/3})$, therefore the conclusion holds. The optimal $\text{MSE}_T$ tends to $0$ as $B\to\infty$. The intuition is that given unlimited data ($B\to\infty$) and finite horizon $T$, the approximation error and variance both reduce to $0$ due to $h\to0$ and having infinitely many episodes.
>
> > Figure $3$
>
> `Re:` Figure $3$ was meant to illustrate the comparison between a line fitted on data with a fixed order in $B$, given by the theoretical analysis, and experimental data points. We did not intend to force the results: InvertedDoublePendulum was indeed a negative example. The provided chart was indeed meant to show that the approximation given by the knowledge of the order in $B$ is good, but not always accurate. We will state it more clearly in further revisions to improve clarity.
>
> > "estimation horizon is the same length as the episode"
>
> `Re:` Perhaps we misunderstand the reviewer’s point: we explicitly analyzed the discounted infinite horizon case where the estimation horizon $T$ is necessarily different from the system horizon $\tau$, and so we did not assume the estimation horizon is the same length as the episode.
>
> ### Non-uniform horizon
> This is an interesting issue. We first note that the technical analysis is grounded in LQR systems, where the conventional setting involves a fixed episode horizon, either finite or infinite (with discounting), without considering early termination.\
> In the nonlinear experiments, we utilized the benchmark Mujoco environments, which have a fixed time limit. Coupled with our assumption of a stable policy, this ensures that all episodes maintain consistent lengths. Hence a uniform horizon aligns well with the scope of our work in both the theoretical analysis and experimental results.\
> While we acknowledge the potential interest in extending our analysis to scenarios with non-uniform horizons, this would necessitate a different formulation, potentially adopting the approach proposed in [Poiani et.al 2023]. We would like to explore this direction in future work.
>
> ### Adaptive sampling
> An adaptive sampling procedure, similar to those adopted in solving SDEs [Ilie et. al. 2015], could represent an interesting direction for future work. Nonetheless, there are few factors that complicate its applicability in the current setting.\
> From a practical perspective, the most important difficulty lies in the additional hyperparameters, which needs to be fine-tuned.\
> With regard to the analysis, establishing a theoretical framework for investigating non-uniform step-size is not straightforward, as it involves additional degrees of freedom, due to the fact that the number of steps in each trajectory does not need to be consistent, thus making the optimal step-sizes under such a setting not necessarily unique.
>
> **References:**
>
> Poiani, R., Metelli, A. M., & Restelli, M. (2023). *Truncating Trajectories in Monte Carlo Reinforcement Learning.* arXiv preprint arXiv:2305.04361.
>
> Ilie, S., Jackson, K. R., & Enright, W. H. (2015). *Adaptive time-stepping for the strong numerical solution of stochastic differential equations.* Numerical Algorithms, 68(4), 791-812.

---

> > ### Comment · Reviewer_uBm7 · 2023-08-13
> >
> > Thank you to the authors for responding to my comments and providing further references, it has been useful. Apologies that I wasn't very clear before, my comment about the estimation horizon was also relating to the non-uniform episodes I believe (and therefore addressed in your comments above).
> >
> > I think this is a moderate-to-high impact paper that should be accepted so I will leave my score as 6.

---

### Official Review · Reviewer_37Kf · 2023-07-12

**Soundness:** 3 good
**Presentation:** 3 good
**Contribution:** 3 good
**Rating:** 7
**Confidence:** 3

**Summary:**

The authors study the impact of time discretization on RL methods in order to improve data efficiency and show that that data efficiency can be significantly improved by leveraging a precise understanding of the trade-off between approximation error and statistical estimation error in value estimation. They conduct a theoretical analysis followed by numerical ones on MuJoCo environments to demonstrate value.

**Strengths:**

Quality and Clarity: The paper is well written and presents a clear and detailed theoretical framework to study the effect of time discretization in terms of data efficiency for control problems. The derivations are easy to follow with necessary background provided in the supplementary materials.

Originality: Marginal improvements are demonstrated by providing the necessary theoretical foundation which is missing from some prior work like Lutter et al as pointed out by the authors.

Significance: The paper addresses an important and relevant problem of improving data efficiency for data hungry RL algorithms.


**Weaknesses:**

Sample efficiency is a huge pain point for solving control tasks in real world where collecting large amounts of data is costly, infeasible or even dangerous. A key weakness I found is the fidelity of this approach in real world scenario. I found the theory to be sound, but built on a lot of simplifying assumptions that do not hold true in real world. So although the work is interesting, I am not sure how strong will these results hold on real world data where the noise isn't additive, or the model isn't linear.

**Questions:**

No specific questions. All concerns are addressed in the paper+supplementary materials.

**Limitations:**

Authors have done a good job in stating the limitations that I agree with- (a) it would be interesting to see how this work extends to more advanced techniques that MC estimation, like TD learning. (b) Secondly, studying the full control setting that includes policy optimization and, (c) simplifying assumptions like linear models with additive noise are considered, but in real world these assumptions don't hold where the observations are noisy, and systems are non-linear. So how this analysis extends to real world sample efficiency for RL algorithms is unclear from the work.

---

> ### Author Rebuttal · Authors · 2023-08-09
>
> We would like to thank the reviewer for the time reviewing our work and for providing valuable feedback.
>
> While it is true that the technical analysis holds only in the case of linear dynamics, we explicitly conducted experiments in nonlinear systems to understand if the theoretical findings could still hold when certain assumptions were not met. Note indeed that in the non-linear Mujoco environments, the setting is significantly different from the one considered in the exact analysis. Nonetheless, the general results seem to extend quite remarkably to this more challenging scenario. This suggests that the results may have broader applicability than the specific conditions under which the theoretical analysis was established. \
> Many nonlinear behaviors can be approximated by a high-dimensional linear system, which would nevertheless be bounded by our most general results on $n$-dimensional systems, hinting at the fact that similar trade-offs could characterize nonlinear systems as well, as demonstrated through numerical analysis.
>
> We believe that this work broadens the perspective and provides valuable opportunities for further exploration and development.

---

> > ### Comment · Reviewer_37Kf · 2023-08-20
> >
> > I would like to thank the authors for addressing my questions and concerns. I will keep my score and vote for acceptance.

---

### Official Review · Reviewer_cepd · 2023-07-12

**Soundness:** 3 good
**Presentation:** 3 good
**Contribution:** 2 fair
**Rating:** 6
**Confidence:** 4

**Summary:**

The paper studies the optimal temporal discretization level to achieve the best Mean Square Error (MSE) in continuous value estimation, considering a fixed budget constraint (number of total Monte-Carlo samples). The paper provides theoretical analysis on a 1-dimensional Langevin dynamical system with quadratic costs and derives the asymptotic optimal step-size under both finite and infinite horizon settings. Extensive numerical studies, including non-linear MuJoCo environments, support the theoretical findings and reveal the bias-variance tradeoff associated with the choice of temporal discretization level.

**Strengths:**

Motivation and clarity:
- The paper presents a clear motivation and an interesting problem setup regarding temporal resolution in continuous value estimation.
- The paper is well-written with a clear structure. Key messages, such as the problem setup, objectives, key theoretical results, and numerical result figures, are easily understandable.

Quality of the paper:
- Section 3 delves deep into a special case of a 1-dimensional Langevin Process, fully characterizing the MSE and providing insights into the optimal step-size.
- Section 4 builds upon the results from Section 3 and provides extensive numerical studies in various environments.
- The conclusion of the paper effectively demonstrates the limitations and shows a deep understanding of the topic, as well as potential avenues for future research.

**Weaknesses:**

Limitation of the scientific impact:
- My major concern lies in the limitation of the impact, as also mentioned in the paper's conclusion. The optimal resolution results are constrained to Naive Monte-Carlo estimation, which typically serves as a baseline for policy evaluation algorithms and cannot be directly extended to advanced methods such as temporal difference.
- Although the numerical results demonstrate the bias-variance tradeoff for non-linear systems, the current theoretical results on the 1-dimensional linear dynamical system cannot be directly extended to more complex systems, where exact characterization no longer exists.

**Questions:**

Suggestions:
1. If simplicity is not our primary criterion, it would be interesting to explore alternative discretization plans beyond uniform discretization.

---

> ### Author Rebuttal · Authors · 2023-08-10
>
> We would like to thank the reviewer for the time reviewing our work and for providing valuable feedback.
>
> One clarification we would like to respectfully add is that tight bounds for the general case of a linear $n$-dimensional system are established in the paper. That is, the theoretical results are not restricted to the $1$-dimensional case.
>
> ### Monte-Carlo estimation
> While focusing on the case of Monte-Carlo estimation, the analysis serves as an essential starting point for studying the fundamental problem of step-size selection, which has been largely overlooked in the literature and there are currently few works that address this issue - as corroborated by two other reviewers. Historically, the Monte Carlo method has been instrumental in developing other reinforcement learning (RL) algorithms, such as Temporal Difference learning.\
> We agree that this line of research can be further explored by considering different estimators (e.g. system identification case - pointed out by another reviewer - or Temporal Difference estimation) and/or different sampling schemes (e.g. adaptive step-size). In particular, the Temporal Difference estimator has a sufficiently simple form that appears to admit tractable analysis. We are indeed exploring similar analyses for TD and other algorithms as part of our future work.
>
> ### Alternative discretization plans beyond uniform discretization
> It is indeed interesting to consider non-uniform discretization schemes, such as through an adaptive sampling scheme, similar to those adopted in solving SDEs [Ilie et al. 2015]. We view this paper as providing the necessary first steps to providing a basis for evaluating alternative discretization schemes.\
> From a theoretical perspective, a non-uniform step-size creates additional issues, since the number of steps in each trajectory might vary. A deeper investigation of adaptive step-size schemes represents an interesting future direction.
>
> ### Theoretical results for more complex systems
> It is true that the detailed analysis given in the paper requires the exact solution of the stochastic differential equation, which is unavailable for most nonlinear systems. However, one can leverage the tight bounds established for an $n$-dimensional linear system and expand the dimensionality in order to approximate more complex nonlinear dynamics in many cases.\
> This approximation would then exhibit the same trade-off.
>
> We believe that these findings already make a valuable contribution to the field. That said, we agree with the value of further exploration in more complex settings. We thank the reviewer for pointing out these other potential research directions.
>
> **Reference:**
>
> Ilie, S., Jackson, K. R., & Enright, W. H. (2015). *Adaptive time-stepping for the strong numerical solution of stochastic differential equations.* Numerical Algorithms, 68(4), 791-812.

---

### Official Review · Reviewer_hJmD · 2023-08-01

**Soundness:** 4 excellent
**Presentation:** 4 excellent
**Contribution:** 3 good
**Rating:** 7
**Confidence:** 3

**Summary:**

This paper examines the time discretization for continuous value estimation. By analyzing Monte-Carlo value estimation for LQR systems for both finite-horizon and infinite discounted horizon settings, the authors finds that there is a fundamental trade-off between approximation error and statistical error in value estimation, which indicates there is an optimal choice for time discretization that depends on the data budget. The authors also demonstrate the trade-off in numerical simulation of LQR instances and non-linear mujoco environments.



**Strengths:**


- The one dimension example presented in the paper is helpful for understanding
- The experiments in the paper clearly support the theoretical analysis and results.


**Weaknesses:**

- My only concern is the practise of the setting or scenarios: people may argue that the $\delta t$ is not randomly chosen and might be fixed or determined by the system itself. Can the authors illustrate some realistic scenarios that we can leverage the methods you proposed.

**Questions:**

My questions is the weakness point I listed.

---

> ### Author Rebuttal · Authors · 2023-08-10
>
> We would like to thank the reviewer for the time reviewing our work and for providing valuable feedback.
>
> We are motivated by real-world scenarios where one must choose the frequency at which sensors operate to get samples of the system signal. Indeed, in many real-world applications, sensors sample at a higher frequency than necessary, and data is downsampled later on.
>
> A realistic scenario where a practitioner chooses $h$ when evaluating the policy of an in production recommender systems at large streaming companies, such as YouTube or Netflix, where the recommendations are entire homepages on the streaming platform. In this setting, it is infeasible to log every single interaction the system has with each user, due to the massive amount of interactions. The solution adopted in practice is to store only snapshots – subsets – of the latter, i.e. only a fraction of the total interactions and corresponding recommendations are logged. Storing a snapshot is costly, both in memory and in computation, as it interferes with other components of the production pipeline. A high sampling frequency causes real performance issues in the production pipeline and requires a large memory, only to marginally improve the estimated value of the recommender system. This is not desirable if the cost of data outweighs the latter improvement. Our analysis suggests that careful consideration should be taken when setting the snapshot frequency in order to mitigate the adverse effects of having to deal with a cumbersome amount of data.
>
> Another example is when applying RL to water treatment systems [Chen et al. 2021], one is faced with choosing a proper sampling step-size - or sampling frequency of the sensor. Sensors that gather data at a higher frequency tend to be more expensive. Our result suggests that opting for the faster sensor without careful consideration may not be the most economical or efficient strategy.
> Even in scenarios where $h$ is fixed (e.g. when a sensor is already in place in an older system), our result provides insights into whether data efficiency can be further improved in existing systems. The implications are two-fold. First, our findings can identify potential areas for improvement to guide future system upgrades. Second, if $h$ is smaller than the optimal value suggested by our method (i.e., the operating frequency of the sensor is too high), it is not necessary to store all samples.
> Nonetheless, for many sensors used in real-world applications, it is indeed possible to decrease the operating frequency. This insight is particularly useful for edge devices with limited hardware capabilities for storage and computation.
>
> In broad terms, our analysis seeks to understand the associated benefits and costs – with the constraint being a fixed data budget – when downsampling a system.
>
> **Reference:**
>
> Chen, K., Wang, H., Valverde-Perez, B., Zhai, S., Vezzaro, L., Wang, A.  (2021).  _Optimal control towards sustainable wastewater treatment plants based on multi-agent reinforcement learning._  Chemosphere, Volume 279, 130498.

---

> > ### Comment · Reviewer_hJmD · 2023-08-20
> > **Thank authors for the response**
> >
> > I would like to thank the authors for answering my questions and clarifying my concerns. I will keep my score and vote for acceptance.

---

### Decision · Program_Chairs · 2023-09-21

**Decision:**

Accept (poster)

**Comment:**

This paper provides a precise characterization of the approximation, estimation, and truncation errors incurred by Monte-Carlo policy evaluation in continuous-time linear stochastic dynamical systems with quadratic cost. While the setting is well-known, the analysis reveals the fundamental bias-variance trade-off, which is new. All the reviewers reached the consensus that the paper should be accepted. I recommend the paper be accepted. However, the authors should incorporate the suggestions from the reviewers as promised.